SciPost Physics

Submission

# Quenches in initially coupled Tomonaga-Luttinger Liquids: a conformal field theory approach

Paola Ruggiero[1*], Pasquale Calabrese[2,3], Laura Foini[4], Thierry Giamarchi[1],

**1** DQMP, University of Geneva, 24 Quai Ernest-Ansermet, CH-1211 Geneva, Switzerland
**2** SISSA and INFN, Sezione di Trieste, via Bonomea 265, I-34136, Trieste, Italy
**3** International Centre for Theoretical Physics (ICTP), I-34151, Trieste, Italy
**4** IPhT, CNRS, CEA, Université Paris Saclay, 91191 Gif-sur-Yvette, France *
paola.ruggiero@unige.ch

March 28, 2021

## Abstract

We study the quantum quench in two coupled Tomonaga-Luttinger Liquids (TLLs), from the off-critical to the critical regime, relying on the conformal field theory approach and the known solutions for single TLLs. We exploit the factorization of the initial state in terms of a massive and massless modes which emerges in the low energy limit, and we encode the non-equilibrium dynamics in a proper rescaling of the time. In this way, we compute several correlation functions, which at leading order factorize into multipoint functions evaluated at different times for the two modes. Depending on the observable, the contribution from the massive or from the massless mode can be the dominant one, giving rise to exponential or power-law decay in time, respectively. Our results find a direct application in all the quench problems where, in the scaling limit, there are two independent massless fields: these include the Hubbard model, the Gaudin-Yang gas, and tunnel-coupled tubes in cold atoms experiments.

# 1 Introduction

In recent times the theoretical understanding of out-of-equilibrium homogeneous systems in 1D has become central in statistical and condensed matter physics, as counterpart to the enormous experimental advances brought by cold atoms [1, 2]. Several aspects have indeed been tackled by a variety of techniques, ranging from numerical methods (with particular reference to TEBD – time evolved block decimation [3]– and to DRMG – density matrix renormalization group– [4, 5], and its time dependent extension [6, 7]) to field theoretical techniques [8–11], integrability [12–19] and much more (see, e.g., [20–25] as more comprehensive reviews).

Due to the complexity of generic out-of-equilibrium protocols, many studies focused on the simplified setup where the system is prepared in the ground state of some hamiltonian $H_0$ and then let evolve with a different one $H$: the famous *quantum quench*. While bringing important simplifications from a theory viewpoint, following the first remarkable example of Ref. [26], quenches have been realized in a variety of cases in cold atomic systems (see e.g. [27–32]). The possibility of experimental realizations triggered a corresponding theoretical effort to set a framework in which to study quenches [33–37].

While for free models, both on the lattice and in the continuum, quantum quench problems are most often analytically treatable as, e.g., reviewed in [15], it has been understood that the powerful tools of integrability out-of-equilibrium [13, 14] can lead to exact analytic results only for a limited (but very interesting and experimentally relevant)

class of initial states compatible with integrability [38]. Consequently, many interesting scenarios can be studied only with the help some approximate methods.

In this respect, when $H$ is at or close to a quantum critical point, a very powerful approach is brought by conformal methods. Specifically, when the initial hamiltonian is massive, the problem can be tackled relying on an imaginary time path integral approach. In particular, in (1+1)D, the problem is mapped to a boundary conformal field theory (BCFT) one. This is the key result of the works by Calabrese and Cardy [33, 35]. This description gives rise to exponential decay in time of correlations (with decay rate fixed by the initial mass). In contrast, when also the initial hamiltonian is massless, the correlations are expected to decay algebraically. Such behavior is recovered for generic systems relying on the Tomonaga Luttinger Liquid (TLL) paradigm [8, 39, 40], where initial and final hamiltonians are fully characterised by two parameters, known as sound velocity $u$ and Luttinger parameter $K$. In this case, the power-law decays can be related in a simple way to initial and final Luttinger parameters only, as shown by Cazalilla in [34] (see also [9, 41–56] for some generalizations).

These results are somehow complementary, giving access to the dynamics after a quench starting from different classes of initial states. Still, it is important to keep in mind their range of applicability, especially when aiming at describing quantitatively the non-equilibrium dynamics of realistic microscopical models such as spin chains and quantum gases. In fact, at equilibrium it is very well established that the TLL approximation is quantitatively correct in the low-energy/large-distance regime; conversely because of the instantaneous nature of the quench after a quantum quench the system has an energy in the middle of the many-body spectrum and by no means a low-energy description is justified. Consequently, the extent to which Tomonaga Luttinger liquid theory can be used for the quantitative analysis of quench dynamics is a non trivial question. Of course, if the quench is near instantaneous but slow enough compared to high energy scales of the microscopic model (see e.g. [57]), the field theory description remains valid.

We will assume that we can fully describe the system by an effective field theory. In addition to the practical possibility of finding ramps with the proper speed for the field theory to remain valid, the theoretical study of the non-equilibrium dynamics of these conformal systems has its per se interest and provides very fundamental qualitative features that are difficult to get by other means in such generality (e.g., the previously mentioned exponential and power-law decays of correlations). Furthermore, many works attempted a detailed comparison between the CFT predictions and the actual non-equilibrium dynamics of lattice models, as e.g., [58–67] and, maybe surprisingly, it turned out that many features of the quench dynamics are not only qualitatively but also quantitatively captured by the TLL approximation. The effect of perturbations away from TLL model has also been analysed in [68–71] using renormalization group (RG) methods showing that the above mentioned studies provided a useful starting point.

Another crucial point is that the above-mentioned standard TLL approaches are only valid when the system under consideration consists of a *single* field or, trivially, of several independent (pre- and post-quenches) modes. On the other hand, in realistic (even 1D) systems this is not always the case: fermions usually carry spin, thus doubling the degrees of freedom, like in the celebrated (Fermi-)Hubbard model [8, 72] as well as Gaudin-Yang gases [73, 74]; more generally speaking, it is very common to end up in situations where the system at low-energy consists few coupled *species* of quasiparticles like in the experimentally relevant example of two (or more) tubes of interacting cold bosonic gases which are tunnel-coupled [75–83].

However, in the presence of more *interacting species*, the quench problem becomes very challenging both numerically and analytically. As examples, we mention the numer-

ical analysis performed in Refs. [84–94] with a variety of methods. The situation is even more complicated on the analytical side, where only few exact results are available and mainly focused on the characterization of the final steady state [95–97] or work in some limits/approximations [98–100]. It is thus clear that any semi–quantitative, or even qualitative general picture for such problems would be not only useful, but highly desirable and this is the strategy employed in some related works appeared in the literature [101–107]. A great simplification occurs when the Hamiltonian has some symmetry and the problem with two degrees of freedom can be studied by introducing a suitable change of variable leading to effectively decoupled modes. In the case of tunnel coupled condensates this has allowed to study the quench of two identical tubes with a mass coupling between the two that is suddenly removed [104, 108–113].

The study of the problem is, instead, much more complicated when no such obvious change of variable allows to reduce the problem to two decoupled modes, see for instance [105]. In this direction, few recent studies investigated the quench dynamics in the tunnelling coupling of two TLLs with different sound velocity and/or Luttinger parameter [114–116], aiming at understanding the effect of such "imbalance" between them. Using a semiclassical approximation, the problem was solved via a Bogoliubov approach. This approximation gives access to a very rich phenomenology [116], with (i) the emergence of multiple lightcones, separating different decaying regimes; (ii) a *prethermal* regime eventually decaying into a *quasi-thermal* one; (iii) non-trivial effects of a non-zero temperature in the initial state.

However, one may wonder how general the obtained results are. In this work we aim to address this question. Anticipating our main result, we will show that general results can be obtained by relying on conformal methods for a quite large class of initial states (namely, those whose low-energy limit is of the form (28) below).

The paper is organized as follows. In Section 2 we set the problem, including some reminders of the results obtained in Ref. [116] within the semiclassical approximation. In Section 3 we give a brief overview of the two approaches to quantum quenches in conformal field theory for a single field. The problem of two initially coupled TLLs is then studied relying on such results in Section 4 and Section 5. Specifically, Section 4 discusses the factorization of the initial state in a suitable basis, taking place in great generality at low energy, and the decoupling of operators, in the same basis. These are the main ingredients to achieve the calculation of correlation functions, discussed in Section 5. Our main results are Eqs. (37-41) and Eqs. (43-45) for the one- and two-point correlation functions of vertex operators of symmetric and antisymmetric sectors (see below for proper definitions), together with Eqs. (52-53) and Eqs. (54-55) for the correlations of density and current of the initial fields. As it will be clear, however, any correlation function can be computed in a similar fashion. We finally conclude in Section 6. In order to keep the paper fluid to read, we chose to collect most of the calculations in four appendices.

## 2 Setting of the problem

As mentioned in the introduction we are interested in studying the time evolution after a quench in which the post-quench low-energy physics is captured by two *different* Tomonaga-Luttinger liquids (in the CFT language, two free compact bosons). Without loss of generality we can write the post-quench hamiltonian as

$$H = H_{u_1,K_1}[\theta_1, n_1] + H_{u_2,K_2}[\theta_2, n_2] \tag{1}$$

with ($i = 1, 2$)

$$H_{u_i, K_i}[\theta_i, n_i] = \frac{u_i}{2\pi} \int \mathrm{d}x \left[ K_i (\nabla \theta_i)^2 + \frac{1}{K_i} (\pi n_i)^2 \right],\tag{2}$$

with $[n_i(x), \theta_j(x')] = i\hbar\delta(x - x')\delta_{ij}$, and $\{u_i, K_i\}$ the associated speeds of sound and TLL parameters. A possible quadratic coupling between the two modes $i = 1, 2$ can be easily reabsorbed with a canonical transformation and hence we do not write it here.

We assume that the initial state couples the two Hilbert spaces associated to $i = 1, 2$. Otherwise, the problem factorizes in the initial variables and we go back to the dynamics of (two decoupled) single fields, which, as mentioned in the introduction, has been already largely addressed. To have in mind a practical example, the initial state can be thought to be the ground state of

$$H_0 = H + \frac{g}{4\pi} \int dx \, (\theta_1(x) - \theta_2(x))^2 \ .\tag{3}$$

This is exactly the case that three of us considered by Bogoliubov approach in [116] (see also the next subsection). In addition to this explicit form we will discuss in Sec. 2.2 more general initial states that are also captured by our CFT approach.

For each $i = 1, 2$, the Hamiltonian $H_i$ in Eq. (2) can be brought in a diagonal form

$$H_{u_i, K_i} = \sum_p u_i |p| b_{i,p}^\dagger b_{i,p} \,.\tag{4}$$

The fields $\theta_i, n_i$ in (2) are related to the bosonic creation/annihilation operators $b_{i,p}^{(\dagger)}$ via:

$$\theta_i(x) = \frac{i}{\sqrt{L}} \sum_{p \neq 0} e^{-ipx - p/\Lambda} \sqrt{\frac{\pi}{2K_i|p|}} (b_{i,p}^\dagger - b_{i,-p}) + \frac{1}{\sqrt{L}} \theta_{i,0},\tag{5}$$

$$n_i(x) = \frac{1}{\sqrt{L}} \sum_{p \neq 0} e^{-ipx - p/\Lambda} \sqrt{\frac{|p|K_i}{2\pi}} (b_{i,p}^\dagger + b_{i,-p}) + \frac{1}{\sqrt{L}} n_{i,0},\tag{6}$$

where $L$ is the system size, and we introduced an ultraviolet cutoff $\Lambda$ in order to ensure convergence of correlation functions. In the rest of the paper we focus on the thermodynamic limit (TDL), namely infinite system size.

In the following sections we discuss the dynamics of the symmetric and antisymmetric modes $\theta_\pm = (\theta_1 \pm \theta_2)/\sqrt{2}$, which are often relevant in systems with two types of degrees of freedom, as the Hubbard model or the tunnel-coupled condensates. While these variables are obviously the most appropriate ones in the case of two identical systems, as the initial and final Hamiltonian are decoupled in this basis and the quench occurs only in the antisymmetric sector [117, 118], this is not true in general when the two systems are different.

## 2.1 Reminders of Bogoliubov approach and some notation

In the Bogoliubov approach to the quench dynamics [15], the initial state is assumed to be the ground state (or even another eigenstate [119], but we do not consider this case here) of a quadratic hamiltonian, such as the one in (3). Hence, the standard way to solve the quench dynamics of interest is to exploit this quadratic nature and perform a Bogoliubov transformation to diagonalize the initial Hamiltonian as

$$H_0 = \sum_p \lambda_{m,p} \eta_{m,p}^\dagger \eta_{m,p} + \sum_p \lambda_{0,p} \eta_{0,p}^\dagger \eta_{0,p} \ .\tag{7}$$

Here we are interested in theories that have a massive $(m)$ and a massless $(0)$ mode (as for Eq. (3), see [116] for details), i.e. in which the small momentum behavior of the two dispersions reads

$$\lambda_{m,p} = m_0, \qquad\qquad \lambda_{0,p} = v|p|. \tag{8}$$

For the specific case of the hamiltonian (3), considered in Ref. [116], we have $m_0 = \sqrt{\frac{gu}{K}}$ which is the mass of the massive mode in the initial state and $v = \sqrt{\frac{u_1 u_2}{K_1 K_2}} K$ the speed of sound of the orthogonal (massless) mode. The speed of sound $u$ and TLL parameter $K$ of post-quench symmetric (i.e., $\theta_+$) and antisymmetric (i.e., $\theta_-$) modes are equal and are given by

$$uK = \frac{1}{2}(u_1 K_1 + u_2 K_2), \qquad\qquad \frac{u}{K} = \frac{1}{2}\left(\frac{u_1}{K_1} + \frac{u_2}{K_2}\right). \tag{9}$$

Finally, an important role in the following is played by the parameters

$$\Gamma = \frac{uK}{v}, \quad K_+ = \frac{K_1 + K_2}{2}. \tag{10}$$

## 2.2  Quenches from a general class of initial states

In this subsection we discuss the class of initial states to which our methods applies.

In general, within the TLL approximation, the initial state (as e.g. the ground state of (3)) can be factorized as the product of the vacua of the pre-quench modes (the $\eta$ operators of Eq. (7), i.e., $|\psi_0\rangle = |0_{\eta_m}\rangle \otimes |0_{\eta_0}\rangle$, where the states $|0_{\eta_{m/0}}\rangle$ satisfy $\eta_{m/0,p}|0_{\eta_{m/0}}\rangle = 0, \forall p$). However, the non-equilibrium dynamics is accessible in the basis of the post-quench hamiltonian, i.e. in the Fock space generated by the $b_{i,p}^\dagger$ operators in Eq. (4). Because of the quadratic nature of both pre- and post-quench hamiltonians at low energy, the initial state is (up to normalization factor $\mathcal{N}$) a squeezed state of the form

$$|\psi_0\rangle = \frac{1}{\mathcal{N}} \prod_p e^{(b_{1,p}^\dagger, b_{2,p}^\dagger) \mathbf{W}_p \begin{pmatrix} b_{1,-p}^\dagger \\ b_{2,-p}^\dagger \end{pmatrix}} |0\rangle, \tag{11}$$

where $|0\rangle$ is the vacuum of the final Hamiltonian $H$ and $\mathbf{W}_p$ is a $2\times 2$ matrix. While for any specific problem of two coupled TLLs, the precise form of $\mathbf{W}_p$ can be explicitly worked out (see e.g. Ref. [116] for the ground state of (3)), this is not necessary for our aims, since we will need only its small momentum expansion. In fact, in the spirit of RG, we will expand as function of $p$ the matrix elements of $\mathbf{W}_p$ at the leading non-trivial orders and discuss the effects of the neglected terms. Whether a squeezed state approximation is justified in some regimes for general interacting theories is subject of current research [120–124]. To our aims, it is sufficient that such state emerges in the low momentum expansion.

To be concrete, we provide an example, maybe trivial, of the possible initial states. This is the ground state of the most general quadratic hamiltonian in the fields $\theta_i$, when the problem can be recast in hamiltonian (3), at the price of having renormalized Luttinger parameters. Specifically, suppose we consider as initial Hamiltonian

$$H = H_{u_1, K_1}[\theta_1, n_1] + H_{u_2, K_2}[\theta_2, n_2] - \frac{g}{4\pi} \int dx (\alpha \theta_1 - \beta \theta_2)^2, \tag{12}$$

with $\alpha, \beta$ arbitrary real numbers. Then, with the rescaling $\tilde{\theta}_1 = \alpha \theta_1, \tilde{n}_1 = n_1/\alpha, \tilde{\theta}_2 = \beta \theta_2, \tilde{n}_2 = n_2/\beta$, we can rewrite the Hamiltonian as

$$H = H_{u_1, K_1/\alpha^2}[\tilde{\theta}_1, \tilde{n}_1] + H_{u_2, K_2/\beta^2}[\tilde{\theta}_2, \tilde{n}_2] - \frac{g}{4\pi} \int dx (\tilde{\theta}_1 - \tilde{\theta}_2)^2, \tag{13}$$

which is of the form given in (3).

# 3   Preliminary results: quenches in a single CFT

Before studying the setting of two coupled CFTs introduced above, we shorty review the results available for the simpler case of a quench in a single CFT. In this case, the post-quench Hamiltonian is $H_{u,K}[\theta, n]$, while the initial state can be massive, e.g. the ground state of $H_{u,K}[\theta, n] + g/(4\pi) \int \mathrm{d}x\ \theta^2(x)$, or massless, namely the ground state of $H_{u_0,K_0}[\theta, n]$.

## 3.1   Massive quench

This quench has been solved exploiting the conformal invariance of the problem, considering an imaginary time path integral approach, that we recall in this section. The results of this method have been developed in [33,35], and later clarified and generalised in [125–130]. This framework is quite general and applies to quenches starting from a translationally invariant state $|\psi_0\rangle$ with short-range correlations, such as, for example, the ground state of the gapped hamiltonian that we mentioned above.

The objects of interest are expectation values of local operators $\phi_j(x_j)$ after the quench, namely

$$\langle\psi_0|\phi_1(x_1, t_1)\cdots\phi_n(x_n, t_n)|\psi_0\rangle. \tag{14}$$

In imaginary time, Eq. (14) can be represented as a path integral over a strip with operator insertions and $|\psi_0\rangle$ playing the role of boundary condition imposed at initial and final times.

A crucial point is that, exploiting the powerful tools of Renormalization Group (RG) theory of boundary critical phenomena [131], a short-ranged initial state $|\psi_0\rangle$ can be always replaced by the appropriate RG-invariant boundary state $|B\rangle$ to which it flows. The distance of the actual boundary state is taken into account (to leading order) introducing an extrapolation length $\tau_0$, and approximating the state as $|\psi_0\rangle \simeq e^{-\tau_0 H}|B\rangle$, with $H$ the post-quench conformal hamiltonian. Note for later reference that in terms of the creation operators $b_p^\dagger$, it takes the form of a squeezed state,

$$|B\rangle \simeq \prod_{p>0} e^{b_p^\dagger b_{-p}^\dagger}|0_b\rangle \qquad \text{and} \qquad |\psi_0\rangle \simeq \prod_{p>0} e^{(1-2\tau_0 u|p|)\, b_p^\dagger b_{-p}^\dagger}|0_b\rangle, \tag{15}$$

where $|0_b\rangle$ is the vacuum of bosons of the final Hamiltonian (incidentally, squeezed states enter as effective initial states in several different quench contexts, see e.g. [122,132,133]). The extrapolation length $\tau_0$ is expected to be of the order of the inverse gap, e.g., $\tau_0 \propto 1/\sqrt{g}$ in the case of pre-quench hamiltonian (3).

Accordingly, Eq. (14) in can be rewritten as

$$\langle B|\phi_1(x_1, \tau_1)\cdots\phi_n(x_n, \tau_n)|B\rangle, \tag{16}$$

where the problem has been mapped to a boundary conformal field theory (BCFT). Namely, Eq. (16) is given by a path integral over a strip of width $2\tau_0$ with conformally invariant boundary conditions, and operators inserted at $\tau_j$ (with $\tau_j \in [0, 2\tau_0]$). Eq. (16) is often denoted as $\langle\phi_1(x_1, \tau_1)\cdots\phi_n(x_n, \tau_n)\rangle_{\mathrm{slab}(2\tau_0)}$: we will use this convention in Appendix A. One can then rely on standard CFT calculations, based on conformal maps and on the transformation of operators under those, to compute (16) exactly. The imaginary times $\tau_j$ have to be analytically continued to $\tau_0 + it_j$ as the final step, to recover the real time evolution.

Within this framework, very general results can be obtained for $n$-point correlation functions, which show exponential decay in time before relaxation, with the appearance of the famous lightcone effect [33,35,134–139]. The steady state shows a finite correlation

length typical of a thermal system and the deviations from a thermal state generated by the integrability of the model are small scale details not captured by the too simplistic approximation of the initial state (the modifications necessary to observe the relaxation to a generalized Gibbs ensemble [140] within this approach have been worked out in [125]).

## 3.2 Massless quench

The path integral approach of the previous subsection does not apply to initial critical states, which are long ranged and therefore would be associated to a diverging extrapolation length (i.e., a vanishing gap). This case has instead been considered in Ref. [34] (see also [68, 69], that we closely follow in terms of notation, and [9] as review on the subject), where the quench dynamics of a TLL after a sudden change of the TLL parameter, say from $K_0$ to $K_f$, is studied via a Bogoliubov approach.

In this case, the initial hamiltonian is diagonal in some operator basis $\eta_p$, and the final one in some other basis $b_p$. They are related by a Bogoliubov transformation

$$\begin{pmatrix} \eta_p \\ \eta_{-p}^\dagger \end{pmatrix} = \begin{pmatrix} \cosh\delta & -\sinh\delta \\ -\sinh\delta & \cosh\delta \end{pmatrix} \begin{pmatrix} b_p \\ b_{-p}^\dagger \end{pmatrix}, \qquad e^{2\delta} = \frac{K_0}{K_f}. \tag{17}$$

Note that this diagonalization also holds when the quench occurs in the sound velocities as well (i.e., for the more general case $\{u_0, K_0\} \to \{u_f, K_f\}$). Then, the ground state of the initial hamiltonian can be written as a squeezed state in the final basis, i.e.,

$$|0_\eta\rangle = \frac{1}{\mathsf{N}} \prod_{p>0} e^{\mathsf{W} b_p^\dagger b_{-p}^\dagger} |0\rangle, \qquad \mathsf{W} = \tanh\delta = 1 - 2\frac{K_f}{K_f + K_0}, \tag{18}$$

where $\mathsf{N} = \prod_{p>0}(1 - \mathsf{W}^2)^{-1/2}$ is the normalization factor. The main difference as compared to the boundary state (15) is that here $|\mathsf{W}| < 1$: this is ultimately responsible for the power-law decay of correlation functions in the state (18) versus the exponential one in (15).

The relaxation is towards a genuine non-equilibrium steady state, namely a generalized Gibbs ensemble [140], determined by the underlying integrability of the model. In fact, the late-time spatial decay is power-law and governed by an exponent that is different from the one that governs asymptotic ground state correlations (i.e., $K_f$). In particular this Luttinger parameter gets renormalized by a function of the ratio $K_0/K_f$ [68, 69], as one might expect from the transformation in (17).

# 4 Initial state and operators' dynamics

In this section we initiate the conformal field theory study of two initially coupled TLL in the setting of Section 2. We discuss first the low energy properties of the initial state, and then the operators dynamics in the Heisenberg picture.

## 4.1 Initial state: low energy factorization

### 4.1.1 Leading order for small momentum

A generic state of the form (11) cannot be directly handled with conformal methods because of the non-trivial dependence of the momentum in $\mathbf{W}_p$. However, since we already invoked RG ideas to use the TLL approximation, it is natural to focus on the low energy features of the state (11) and take the limit $p \to 0$ of $\mathbf{W}_p$. We consider a zero-momentum

matrix with a massive and a massless mode that can be parametrized (omitting redundant phases) as follows

$$W_0 \equiv \mathbf{W}_{p=0} = \cos^2 \varphi \; \mathbb{I} - \sin^2 \varphi \; S_-(\nu), \qquad S_\pm(\nu) = \begin{pmatrix} \pm \cos 2\nu & \mp \sin 2\nu \\ \sin 2\nu & \cos 2\nu \end{pmatrix}, \qquad (19)$$

where $\mathbb{I}$ is the identity matrix. For example, in the specific case of the initial state being the ground state of the hamiltonian (3), $\varphi$ and $\nu$ are such that

$$\sin^2 \varphi = \frac{K_+}{\Gamma + K_+}, \qquad \nu = \mathrm{atan} \sqrt{\frac{K_1}{K_2}}. \qquad (20)$$

The matrix $W_0$ can be diagonalized via a simple rotation

$$\begin{aligned} b_A^\dagger &= \cos \nu \; b_1^\dagger - \sin \nu \; b_2^\dagger, \\ b_B^\dagger &= \sin \nu \; b_1^\dagger + \cos \nu \; b_2^\dagger \;. \end{aligned} \qquad (21)$$

The two eigenvalues of $W_0$ are $\{1, \cos 2\varphi\}$, associated respectively to two orthogonal sectors that we dub $\{A, B\}$, and the value of these eigenvalues determines the spectrum and the decay of correlations of the two modes, as can be understood from (105). For the Hamiltonian (3), in the case of two identical systems $u_1 = u_2$ and $K_1 = K_2$, the two modes are associated to the antisymmetric/symmetric fields $\theta_\pm$ introduced above. However such correspondence does not hold in general.

The state (11) in the low-energy approximation is thus factorized in the basis $\{A, B\}$. It consists of an infinite mass state (associated to the eigenvalue 1, see Eq. (15)) in the $A$-sector, and a massless state (associated to $|\cos 2\varphi| < 1$, see Eq. (18)) in the $B$-sector. Importantly, these two are the low energy states that characterize the dynamics studied, respectively, by Calabrese-Cardy [33, 35], and by Cazalilla [34], as discussed in Sections 3.1 and 3.2. The dynamics in the $B$-sector can be interpreted as a quench in the TLL parameter. Indeed, for the ground state of the hamiltonian (3), it corresponds to a quench from $\Gamma$ in the initial state to $K_+$ in the post-quench hamiltonian, as follows from identifying in Eq. (18) $\tanh \delta$ with $\cos 2\varphi = \frac{\Gamma - K_+}{\Gamma + K_+}$.

Crucially, the factorization of the state at this order is, by construction, independent of the momentum $p$, in such a way that we can define the fields

$$\begin{cases} \theta_A = \frac{1}{\sqrt{K_A}} \left( \cos \nu \sqrt{K_1} \theta_1 - \sin \nu \sqrt{K_2} \theta_2 \right) \\ \theta_B = \frac{1}{\sqrt{K_B}} \left( \sin \nu \sqrt{K_1} \theta_1 + \cos \nu \sqrt{K_2} \theta_2 \right) \end{cases} \begin{cases} n_A = \sqrt{K_A} (\cos \nu \frac{n_1}{\sqrt{K_1}} - \sin \nu \frac{n_2}{\sqrt{K_2}}) \\ n_B = \sqrt{K_B} (\sin \nu \frac{n_1}{\sqrt{K_1}} + \cos \nu \frac{n_2}{\sqrt{K_2}}) \end{cases}. \qquad (22)$$

The TLL parameters $K_A$ and $K_B$ are auxiliary variables which are free variables and, as we will see, they will not enter in the formulas for the dynamics of physical observables. Always to have in mind a specific example, we can plug in the above equations the value of $\nu$ in Eq. (20), corresponding to hamiltonian (3), to have

$$\theta_A = \sqrt{\frac{K_1 K_2}{2 K_+ K_A}} \left( \theta_1 - \theta_2 \right), \quad \theta_B = \frac{1}{\sqrt{2 K_+ K_B}} \left( K_1 \theta_1 + K_2 \theta_2 \right). \qquad (23)$$

This equations shows that the field $\theta_A$ remains aligned to the massive hamiltonian term $(\theta_1 - \theta_2)$, cfr. Eq. (3).

By inverting Eq. (22) for $\{\theta_i, n_i\}$, and plugging them into the post-quench TLL hamiltonian (1), we get

$$H = H_{u_A, K_A}[\theta_A, n_A] + H_{u_B, K_B}[\theta_B, n_B] +$$
$$+ \lambda_{AB} \left( \sqrt{K_A K_B} \int dx \nabla \theta_A \nabla \theta_B + \frac{\pi^2}{\sqrt{K_A K_B}} \int dx n_A n_B \right), \quad (24)$$

with the coupling of the $A - B$ sectors given by

$$\lambda_{AB} = \frac{(u_1 - u_2)}{\pi} \cos \nu \sin \nu \,, \tag{25}$$

so, in general, the two sectors are coupled. Moreover, we have

$$u_A = u_1 \cos^2 \nu + u_2 \sin^2 \nu, \tag{26}$$

$$u_B = u_1 \sin^2 \nu + u_2 \cos^2 \nu, \tag{27}$$

that fix the sound velocities $u_{A/B}$ of $\theta_{A/B}$. In the case of the ground state of hamiltonian (3), these two velocities are $u_A = \frac{K_1 K_2}{K_+} \frac{u}{K}$ and $u_B = \frac{1}{K_+} uK$.

We conclude this subsection with two comments. The modes $A/B$ allow us to write the initial state as a factorized squeezed state at low-energy with operators acting on the physical vacuum of the post-quench hamiltonian. This is different from writing the state as product of the two pre-quench vacua in Eq. (7). As a little detour, we note that the rotation that diagonalizes $W_0$ in (19) is the same one introduced in Ref. [141] for permeable interfaces in CFT. Indeed, there the scattering matrix is just given by either $S_+(\nu)$ or $S_-(\nu)$. Given that $W_0$ and $S_-$ commute, they are diagonalized by the same transformation. This observation is the starting point for a possible connection between permeable interfaces and quench problems that will be investigated in a forthcoming work [142].

### 4.1.2 Beyond the leading order

The next to leading order in $p$ of the initial state $|\psi_0\rangle$ in general breaks the factorization in the $A/B$ sectors. It is then convenient to write these next-to-leading order terms in $p$ directly in in the basis $A/B$ in which the initial state has the general form

$$|\psi_0\rangle = \frac{1}{\mathcal{N}} \prod_{p>0} e^{(b_{A,p}^\dagger, b_{B,p}^\dagger) W_p^{(1)} \begin{pmatrix} b_{A,-p}^\dagger \\ b_{B,-p}^\dagger \end{pmatrix}} |0\rangle, \qquad W_p^{(1)} = \begin{pmatrix} 1 - 2\tau_A u_A|p| & 2\gamma|p| \\ 2\gamma|p| & \cos 2\varphi(1 - 2\tau_B u_B|p|) \end{pmatrix}, \tag{28}$$

where the normalization $\mathcal{N}$ is reported in Appendix D, see Eq. (110). Clearly $\tau_A, \tau_B$ and $\gamma$ are functions of the initial parameters $\{K_i, u_i\}$, but the precise functional dependence is actually not needed. The two velocities $u_A$ and $u_B$ in $W_p$ are defined in (26) and (27), respectively. $W_p^{(1)}$ is by definition $\mathbf{W}_p$ up to order $O(p)$.

The parameter $\tau_A$ in the $AA$-component in $W_p^{(1)}$ is nothing but the extrapolation length of the massive quench introduced ad hoc in the previous section (cfr. Eq. (15)). This length is interpreted as the "distance" from the infinite-mass state (here as a result of the exact calculation, in agreement with the general expectation from RG arguments in [35]). As already mentioned, it is expected to be of the order of the inverse gap $m_0^{-1}$. This term is the one generating exponential decay of correlation functions.

The $O(p)$ correction to the $BB$-component (parametrised by $\tau_B$), only produce subleading corrections, as shown in Appendix D. Hence it is neglected in what follows.

The factorization of the initial state is spoiled by the presence of the off-diagonal matrix element $\gamma \neq 0$. To proceed, however, in the following we are going to assume a diagonal form of the state also at this order (i.e., $\gamma = 0$). The consequences of a non-zero value of $\gamma$ will be discussed for the correlation functions under consideration. As we are going to argue, the role of $\gamma$ is to renormalize subleading power-law exponents in some cases, or just modify non-universal prefactors in other. Crucially, it will never affect the leading term of the correlation functions of interest.

## 4.2  Decoupling of operator dynamics

We are going to work out the non-equilibrium dynamics in the Heisenberg picture, in which the time dependence is entirely encoded into operators, while the state does not evolve. A suitable rescaling of the times will allow us to always write our observables in a decoupled form with respect to the $A$ and $B$ degrees of freedom.

Let us focus of the field $\theta_1$, for $\theta_2$ the derivation is identical. Its dynamics is given by

$$\theta_1(x,t) = e^{iH_{u_1,K_1}[\theta_1,n_1]t}\theta_1(x)e^{-iH_{u_1,K_1}[\theta_1,n_1]t}. \tag{29}$$

In the above equation we can replace

$$iH_{u_1,K_1}[\theta_1,n_1]\,t \to i\left(H_{u_0,K_1}[\theta_1,n_1] + H_{u_0,K_2}[\theta_2,n_2]\right)\frac{u_1}{u_0}t \tag{30}$$

with $u_0$ a common (arbitrary) velocity for the two TLL hamiltonians of $\theta_{1/2}$. The presence of $H_{u_0,K_2}[\theta_2,n_2]$ does not affect the dynamics of $\theta_1$ because the two commute. Importantly, due to the rescaling of time, now $\theta_{1/2}$ have the same (auxiliary and fictitious) sound velocity $u_0$. Then (cfr. Eq. (24))

$$H_{u_0,K_1}[\theta_1,n_1] + H_{u_0,K_2}[\theta_1,n_1] = H_{u_0,K_A}[\theta_A,n_A] + H_{u_0,K_B}[\theta_B,n_B] \tag{31}$$

where we used that, in these rescaled time, $u_1 = u_2 = u_0$ implies (see Eqs. (26-27)) $u_A = u_B = u_0$ as well, and $\lambda_{AB} = 0$ in (24). This is crucial, because now the hamiltonian in the rhs of (31) acts separately on $\theta_{A/B}$. Finally, defining the rescaled times

$$t_i^a = t\frac{u_i}{u_a}, \qquad i = 1,2 \ \text{ and } \ a = A,B, \tag{32}$$

the rhs of (30) can be recast in the form

$$t_1^A H_{u_A,K_A}[\theta_A,n_A] + t_1^B H_{u_B,K_B}[\theta_B,n_B] \tag{33}$$

which plugged in (29) gives

$$\begin{aligned}
\theta_1(x,t) &= \sqrt{\frac{K_A}{K_1}}\cos\nu\,\theta_A(x,t_1^A) + \sqrt{\frac{K_B}{K_1}}\sin\nu\,\theta_B(x,t_1^B),\\[4pt]
\theta_2(x,t) &= -\sqrt{\frac{K_A}{K_2}}\sin\nu\,\theta_A(x,t_2^A) + \sqrt{\frac{K_B}{K_2}}\cos\nu\,\theta_B(x,t_2^B)\ ,
\end{aligned} \tag{34}$$

where the second equation for $\theta_2$ follows from a very similar calculation. In fact, (34) is nothing but the time-dependent version of Eq. (23).

The rescaling of time introduced above is particularly important when considering observables which are functions of both $\theta_1$ and $\theta_2$, such as, for example, of the symmetric and antisymmetric fields $\theta_\pm$, which decouple in terms of $\theta_{A/B}$ at any time

$$\begin{aligned}
\theta_\pm(x,t) = \\
\frac{1}{2}\Big[\sqrt{K_A}\Big(\frac{\cos\nu}{\sqrt{K_1}}\theta_A(x,t_1^A) \mp \frac{\sin\nu}{\sqrt{K_2}}\theta_A(x,t_2^A)\Big) + \sqrt{K_B}\Big(\frac{\sin\nu}{\sqrt{K_1}}\theta_B(x,t_1^B) \pm \frac{\cos\nu}{\sqrt{K_2}}\theta_B(x,t_2^B)\Big)\Big].
\end{aligned} \tag{35}$$

In summary, the general idea, exploited in the following section, is to use a time rescaling to reabsorb the different velocities of the two initial LLs into the times at which the observables are evaluated. Hence, using rescaled modes with the same sound velocity is enough to ensure an exact decoupling into the time-dependent $\theta_A$ and $\theta_B$ at any time. The price to pay is that equal-time observables and correlations become multi-times ones. This is evident in (35), where a single time in the lhs results in two different times in the rhs.

# 5 Correlation functions

In the previous section, we achieved the two necessary conditions to compute correlations functions, namely

- the factorization of the state in the basis which diagonalizes the fields $\theta_{A/B}$ (assuming to neglect the coupling $\gamma$ in Eq. (28));

- the decoupling of the operator dynamics with respect to the same basis.

Using these properties, all the correlation functions can be computed independently in the massive and in the massless sector as multi-point functions at different times.

In particular, for the massive sector, one can apply the method developed in [33,35], while the computations in the massless sector are equivalent to those in [34]. While it is possible to derive general results valid for an arbitrary initial state of the form (28), the formulas become soon very cumbersome. For this reason, we specialise this entire section to the initial state given by the ground state fo the hamiltonian (3). The correlations for any other choice of the parameters $\varphi$ and $\nu$ in Eq. (19) can be worked out exactly in the same manner. In the following, we will see how our conformal approach provides the correct result of the leading decay of the correlation functions.

## 5.1 Vertex operators

### 5.1.1 One-point functions of $\theta_\pm$

As a first non-trivial example to show the importance of both initial state factorization and time rescaling, we consider the exponential one-point functions of $\theta_\pm$ (i.e., vertex operators in CFT language)

$$C_{1,\gamma}^{\pm}(t) \equiv \left\langle\!\!\left\langle e^{i\sqrt{2}\theta_\pm(t)} \right\rangle\!\!\right\rangle_\gamma , \tag{36}$$

where $\langle\!\langle \cdot \rangle\!\rangle_\gamma$ denotes the expectation value over the state (28), and below we consider $\gamma = 0$. These correlations of $\theta_\pm$ are the experimentally relevant in the context of tunneled-coupled tubes in cold atoms experiments [83]. They are also the most natural also in the Hubbard and Gaudin-Yang models, where they are associated with spin and charge sectors [72].

Making use of the decomposition derived above when $\gamma = 0$ (cfr. Eq. (35)), Eq. (36) can be cast in the following form

$$C_{1,0}^{\pm}(t) = \left\langle e^{i\sqrt{\frac{K_A}{2K_+}}\left(\sqrt{\frac{K_2}{K_1}}\theta_A(t_1)\mp\sqrt{\frac{K_1}{K_2}}\theta_A(t_2)\right)} \right\rangle \left\langle e^{i\sqrt{\frac{K_B}{2K_+}}(\theta_B(t_1)\pm\theta_B(t_2))} \right\rangle , \tag{37}$$

where from now on expectation values of observables which are functions of $\theta_A$ are understood to be taken on the $AA$-component of the state (28). Similarly, functions of $\theta_B$ are evaluated on the $BB$-component of (28) with $\tau_B = 0$ (as already mentioned, it does not contribute up to subleading corrections). Moreover, to lighten the notation, we simply used $t_i$ instead of $t_i^a$, because the correlation univocally specifies whether $a = A$ or $a = B$.

Thus, we conclude that one point functions of exponential of $\theta_\pm$ become a product of *two-point* functions of $\theta_{A/B}$. Now massive and massless part can be computed separately.

We start from the massive part. Using the approach of Section 3.1, the object that we need to evaluate is a path integral over a slab of width $W = 2\tau_A$ with the two vertex operators (which are primary operators in the CFT) inserted at different points. The final result reads (up to a unimportant prefactor)

$$\left\langle e^{i\sqrt{\frac{K_A}{2K_+}}\left(\sqrt{\frac{K_2}{K_1}}\theta_A(t_1)\mp\sqrt{\frac{K_1}{K_2}}\theta_A(t_2)\right)} \right\rangle \simeq e^{-\frac{\pi}{16\tau_A}\frac{K}{u}\left(\frac{1}{K_1 K_2}(\mp|u_1+u_2|\pm|u_1-u_2|)+\frac{1}{K_1^2}|u_1|+\frac{1}{K_2^2}|u_2|\right)t} , \tag{38}$$

where we used Eq. (61) in Appendix A with the specific values $h_1 = \frac{1}{16K_+}\frac{K_2}{K_1}, h_2 = \frac{1}{16K_+}\frac{K_1}{K_2}$ for the conformal weights of the corresponding vertex operators. Eq. (38) reproduces the leading behavior for large $t$ of $C_{1,\gamma}^-(t)$ obtained in Ref. [116] via a Bogoliubov calculation if we identify

$$\tau_A = 1/m_0\,, \tag{39}$$

which is consistent with the standard interpretation of $\tau_A$ as inverse initial mass gap [33].

Let us now move to the $B$ mode. In this sector the initial state is massless, so that we can use the results in [34] for the two-point function, but generalized to the case of unequal times. This is, once again, a standard Bogoliubov calculation (that we report in Appendix B). The final result reads

$$\left\langle e^{i\sqrt{\frac{K_B}{2K_+}}(\theta_B(0,t_1)\pm\theta_B(0,t_2))} \right\rangle = e^{-\frac{K_B}{4K_+}\langle[\theta_B(0,t_1)\pm\theta_B(0,t_2)]^2\rangle} \tag{40}$$

with

$$\langle[\theta_B(0,t_1)\pm\theta_B(0,t_2)]^2\rangle = \frac{1}{2K_B}\int_0^\infty \frac{dp}{p}\times$$

$$\times\left\{\mathbb{K}_+\left[1\pm\cos(p(u_1-u_2)t)\right] - \mathbb{K}_-\left[\frac{1}{2}(\cos(2pu_1t)+\cos(2pu_2t))\pm\cos(p(u_1+u_2)t)\right]\right\} \tag{41}$$

and $\mathbb{K}_\pm = \left(\frac{\Gamma}{K_+}\pm\frac{K_+}{\Gamma}\right)$ (cfr. Eq. (76), specialized to our quench with $K_0/K_f = \Gamma/K_+$). According to the $\pm$ sign, this integral has or has not an infrared (small $p$) divergence. Specifically, such divergence gives $C_{1,0}^+(t) = 0$ when plugging Eq. (41) in (40) (with the $+$ sign). This in fact is the correct result for $C_{1,\gamma}^+(t)$ known also from Ref. [116] (see Eq. (83) in App. C), which remains valid for $\gamma \neq 0$. Conversely, Eq. (40) (together with (41)) leads to a power-law decay for $C_{1,0}^-(t)$. The exponent is however different from the one found for $\gamma \neq 0$ (cfr. Eq. (83) in App. C). We will discuss this discrepancy in Section 5.3. For now, we just point out that at leading order $\log C_{1,\gamma}^\pm(t) = \log C_{1,0}^\pm(t)$, while the same is not guaranteed for subleading corrections (the logarithm is important for the correctness of this statement, since the corrections from the massless sector to $C_{1,\gamma}^\pm$ are multiplicative).

### 5.1.2 Two-point functions of $\theta_\pm$

Similar results can be found for the (exponential) two-point function of $\theta_\pm$

$$C_{2,\gamma}^\pm(x,t) \equiv \left\langle\!\!\left\langle e^{i\sqrt{2}[\theta_\pm(x,t)-\theta_\pm(0,t)]} \right\rangle\!\!\right\rangle_\gamma . \tag{42}$$

As before, we start by rewriting it in a factorized form for $\gamma = 0$, i.e.

$$C_{2,0}^\pm(x,t) = \left\langle e^{i\sqrt{\frac{K_A}{2K_+}}\left[\sqrt{\frac{K_2}{K_1}}(\theta_A(x,t_1)-\theta_A(0,t_1))\mp\sqrt{\frac{K_1}{K_2}}(\theta_A(x,t_2)-\theta_A(0,t_2))\right]}\right\rangle\times$$

$$\times\left\langle e^{i\sqrt{\frac{K_B}{2K_+}}[\theta_B(x,t_1)-\theta_B(0,t_1)\pm(\theta_B(x,t_2)-\theta_B(0,t_2))]}\right\rangle. \tag{43}$$

Therefore, vertex two-point functions of $\theta_\pm$ are mapped into the product of two four-point functions of $\theta_{A/B}$, that we can compute separately.

For the massive part, we now have a four-point function to be evaluated in the same strip geometry considered before. The result is given by (see Appendix A, Eq. (66))

$$
\left\langle e^{i\sqrt{\frac{K_A}{2K_+}}\left[\sqrt{\frac{K_2}{K_1}}(\theta_A(x,t_1)-\theta_A(0,t_1))\mp\sqrt{\frac{K_1}{K_2}}(\theta_A(x,t_2)-\theta_A(0,t_2))\right]}\right\rangle \simeq
$$

$$
\simeq \begin{cases}
e^{-\frac{\pi}{16}\frac{1}{\tau_A}\frac{K}{u}\left[\frac{1}{K_1^2}2u_1t+\frac{1}{K_2^2}2u_2t\mp\frac{2}{K_1K_2}((u_1+u_2)t-|u_1-u_2|t)\right]} & x > 2u_1t \\[2ex]
e^{-\frac{\pi}{16}\frac{1}{\tau_A}\frac{K}{u}\left[\frac{1}{K_1^2}x+\frac{1}{K_2^2}2u_2t\mp\frac{2}{K_1K_2}((u_1+u_2)t-|u_1-u_2|t)\right]} & 2u_1t > x > (u_1+u_2)t \\[2ex]
e^{-\frac{\pi}{16}\frac{1}{\tau_A}\frac{K}{u}\left[\frac{1}{K_1^2}x+\frac{1}{K_2^2}2u_2t\mp\frac{2}{K_1K_2}(x-|u_1-u_2|t)\right]} & (u_1+u_2)t > x > 2u_2t \\[2ex]
e^{-\frac{\pi}{16}\frac{1}{\tau_A}\frac{K}{u}\left[\frac{1}{K_1^2}+\frac{1}{K_2^2}\mp\frac{2}{K_1K_2}\right]x\mp\frac{\pi}{16}\frac{1}{\tau_A}\frac{K}{u}\frac{2}{K_1K_2}|u_1-u_2|t} & 2u_2t > x > |u_1-u_2|t \\[2ex]
e^{-\frac{\pi}{16}\frac{1}{\tau_A}\frac{K}{u}\left[\frac{1}{K_1^2}+\frac{1}{K_2^2}\right]x} & |u_1-u_2|t > x
\end{cases} \tag{44}
$$

where, without loss of generality, we assumed $u_1 > u_2$. Again, if we fix $\tau_A$ as in (39), this reproduces the correct exponential decay of both $C_{2,\gamma}^\pm$ found in [116].

Given that the $B$ sector provides algebraic correlation, the exponential contribution in Eq. (44) represents always the leading term both in $x$ and $t$, as already pointed out in Ref. [116]. A possible special case is the short time regime $2u_1t < x$ (first case in (44)) where there is no $x$-dependence. Hence, the possible space dependence is entirely in the subleading power-law contributions which we now study. The result for the $B$-part of this two-point function (derived in Appendix B) is

$$
\left\langle e^{i\sqrt{\frac{K_B}{2K_+}}\left[(\theta_B(x,t_1)-\theta_B(0,t_1))\pm(\theta_B(x,t_2)-\theta_B(0,t_2))\right]}\right\rangle \sim
$$

$$
\sim |x^2|^{\frac{\mathbb{K}_+}{8K_+}}\left|1-\frac{x^2}{(|u_1-u_2|t)^2}\right|^{\pm\frac{\mathbb{K}_+}{8K_+}}\left|1-\frac{x^2}{(2u_1t)^2}\right|^{-\frac{\mathbb{K}_-}{16K_+}}\left|1-\frac{x^2}{(2u_2t)^2}\right|^{-\frac{\mathbb{K}_-}{16K_+}}\left|1-\frac{x^2}{(|u_1+u_2|t)^2}\right|^{\mp\frac{\mathbb{K}_-}{8K_+}}.
$$
$$\tag{45}$$

Note that from (45) one can easily read off all the different regimes (the same as in Eq. (44)), sharply separated by lightcones. Those are nonetheless smoothen out when reintroducing the ultraviolet cutoff [116]. Note also that both $K_A$ and $K_B$ cancel in the above expressions.

In the aforementioned regime of short time ($x \gg u_1t$), Eq. (45) gives that $C_{2,0}^-$ is constant (the various exponents sum up to zero) and $C_{2,0}^+ \sim |x|^{(K_0/K_f)/(K_+)} = |x|^{1/\Gamma}$. In this regime, these correlation functions match exactly the results from the Bogoliubov calculation in [116] (reported, for completeness, in Appendix C).

In the other regimes, the power-law scaling in Eq. (45) have exponents that are, in general, different compared to the ones for $\gamma \neq 0$. As mentioned already for the one-point functions, this disagreement represent the limits of the conformal method, which does not gives access to *all* power-law contributions. Anyway, the conclusion also for the two-point function, is that the leading term is well captured is all the regimes. We will come back to this issue in Section 5.3.

## 5.2 Derivative operators

We focus here on fluctuations of the initial fields ($i = 1, 2$)

$$
D_{2,\gamma}^{ij}(x,t) \equiv \langle\!\langle n_i(x,t)n_j(0,t)\rangle\!\rangle_\gamma, \quad J_{2,\gamma}^{ij}(x,t) \equiv \langle\!\langle j_i(x,t)j_j(0,t)\rangle\!\rangle_\gamma, \tag{46}
$$

where $j_i(x, t)$ the current density associated to $\theta_i(x, t)$. Density and current correlations can be related to correlators of the derivative operators, which is also a primary operator of the CFT [147]. In fact it holds

$$
\begin{aligned}
D_{2,\gamma}^{ij}(x, t) &= \frac{K_i K_j}{u_i u_j \pi^2} \langle\!\langle \partial_t \theta_i(x, t) \partial_s \theta_j(0, s) \rangle\!\rangle_\gamma |_{s=t}, \\
J_{2,\gamma}^{ij}(x, t) &= \frac{1}{\pi^2} \langle\!\langle \partial_x \theta_i(x, t_i) \partial_y \theta_j(y, t_j) \rangle\!\rangle_\gamma |_{y=0}.
\end{aligned}
\tag{47}
$$

For definiteness, below we look at density-density correlations, while results for the currents can be similarly derived. Following the same logic used for the vertex operators, we exploit the factorization of the state (28) and the decoupling of observable to get

$$
D_{2,0}^{ij}(x, t) = \frac{K_i K_j}{u_i u_j \pi^2} \times
$$
$$
\times \left( (-1)^{i+j} \left( \frac{K_1}{K_2} \right)^{\frac{(-1)^i + (-1)^j}{2}} \frac{K_A}{2K_+} \langle \partial_t \theta_A(x, t_i) \partial_t \theta_A(0, t_j) \rangle + \frac{K_B}{2K_+} \langle \partial_t \theta_B(x, t_i) \partial_t \theta_B(0, t_j) \rangle \right).
\tag{48}
$$

We see that the first term in the above equation is associated to the massive mode, and therefore, according to [33, 35], decays exponentially in time. Hence, the leading term is now given by the part associated to the massless mode, giving rise to a power law decay, according to [34]. Such power-law decay comes out very naturally within this approach.

To get this leading term, we compute

$$
\langle \partial_t \theta_B(x, t_i) \partial_s \theta_B(0, s_j) \rangle = -\frac{1}{2} \partial_t \partial_s \langle [\theta(x, t_i) - \theta(0, s_j)]^2 \rangle
\tag{49}
$$

where we defined $s_j \equiv s_j^a = u_j/u_a s$ with $a = \{1, 2\}$. This is a two-point function at equal times when $i = j$ and at different times when $i \neq j$. In both cases we can evaluate it using (49) and the results in Appendix B. For $i \neq j$ we get

$$
\langle \partial_t \theta_B(x, t_i) \partial_s \theta_B(0, s_j) \rangle|_{s=t} = -\frac{u_i u_j}{4K_+} \times
$$
$$
\left[ \frac{\mathbb{K}_+}{2} \left( \frac{1}{(|u_i - u_j| t + x)^2} + \frac{1}{(|u_i - u_j| t - x)^2} \right) + \frac{\mathbb{K}_-}{2} \left( \frac{1}{(|u_i + u_j| + x)^2} + \frac{1}{(|u_i + u_j| - x)^2} \right) \right],
\tag{50}
$$

and in the case $i = j$

$$
\langle \partial_t \theta_B(x, t_i) \partial_s \theta_B(0, s_i) \rangle|_{s=t} = -\frac{u_i^2}{4} \left( \frac{\mathbb{K}_+}{K_+} \frac{1}{x^2} + \frac{\mathbb{K}_-}{K_+} \frac{1}{2} \left( \frac{1}{(x - 2u_i t)^2} + \frac{1}{(x + 2u_i t)^2} \right) \right).
\tag{51}
$$

The density-density fluctuations finally read ($i \neq j$)

$$
D_{2,0}^{12}(x, t) = -\frac{K_1 K_2}{\pi^2} \frac{1}{8}
\begin{cases}
\frac{\mathbb{K}_+ + \mathbb{K}_-}{K_+} \frac{1}{x^2} & |u_1 + u_2| t \ll x \\[2mm]
\frac{\mathbb{K}_+}{K_+} \frac{1}{t^2 c^2 |u_1 - u_2|^2} + \frac{\mathbb{K}_-}{K_+} \frac{1}{x^2 c^2 |u_1 + u_2|^2} & |u_1 - u_2| t \ll x \ll |u_1 + u_2| t \\[2mm]
\left( \frac{\mathbb{K}_+}{K_+} \frac{1}{|u_1 - u_2|^2} + \frac{\mathbb{K}_-}{K_+} \frac{1}{|u_1 + u_2|^2} \right) \frac{1}{t^2} & x \ll |u_1 - u_2| t
\end{cases}
\tag{52}
$$

where above we defined $c = x/t < \infty$, and (for $i = j$)

$$
D_{2,0}^{ii}(x, t) = -\frac{K_i^2}{\pi^2} \frac{1}{8}
\begin{cases}
\frac{\mathbb{K}_+ + \mathbb{K}_-}{K_+} \frac{1}{x^2} & 2u_i t \ll x \\[2mm]
\frac{\mathbb{K}_+}{K_+} \frac{1}{x^2} + \frac{\mathbb{K}_-}{K_+} \frac{1}{(2u_i t)^2} & x \ll 2u_i t
\end{cases}.
\tag{53}
$$

Similarly, for the current-current correlations we get for $i \neq j$

$$J_{2,0}^{12}(x,t) = -\frac{1}{8\pi^2} \begin{cases} \frac{\mathbb{K}_+ - \mathbb{K}_-}{K_+} \frac{1}{x^2} & |u_1 + u_2|t \ll x \\ \frac{\mathbb{K}_+}{K_+} \frac{1}{t^2 c^2 |u_1 - u_2|^2} - \frac{\mathbb{K}_-}{K_+} \frac{1}{x^2 c^2 |u_1 + u_2|^2} & |u_1 - u_2|t \ll x \ll |u_1 + u_2|t \\ \left( \frac{\mathbb{K}_+}{K_+} \frac{1}{|u_1 - u_2|^2} - \frac{\mathbb{K}_-}{K_+} \frac{1}{|u_1 + u_2|^2} \right) \frac{1}{t^2} & x \ll |u_1 - u_2|t \end{cases} \quad (54)$$

and, for $i = j$,

$$J_{2,0}^{ii}(x,t) = -\frac{1}{8\pi^2} \begin{cases} \frac{\mathbb{K}_+ - \mathbb{K}_-}{K_+} \frac{1}{x^2} & 2u_i t \ll x \\ \frac{\mathbb{K}_+}{K_+} \frac{1}{x^2} - \frac{\mathbb{K}_-}{K_+} \frac{1}{(2u_i t)^2} & x \ll 2u_i t \end{cases}. \quad (55)$$

As we are going to justify in the following subsection, the leading algebraic decay is not influenced by the inclusion of a $\gamma \neq 0$ term.

## 5.3 Corrections from off-diagonal terms and comparison with Bogoliubov approach

In the previous two subsections we calculated several correlation functions following standard RG ideas which are completely under control at equilibrium. In particular, we focused on the first terms in a small momentum expansion of the initial state. However, it in unclear how these RG reasonings capture the real out-of-equilibrium time evolution of the two generally coupled TLLs. The crucial point is represented by the generic form of the initial state in Eq. (28) which shows that at order $O(p)$ (i.e., for $\gamma \neq 0$) a term breaking the factorization of the initial state appears, modifying qualitatively a few aspects of our approach. It is straightforward to realize that this term generates further algebraic decay in space and time separations, as can be understood from the results derived in Appendix D. As a consequence, for all the correlation functions with a leading exponential behavior, the presence of such off-diagonal coupling only provide *subleading* corrections to the result we obtained assuming factorization. Conversely, power-law terms are in generically affected by the presence of the off-diagonal term, and so are not correctly captured within our approach working within a factorized initial state. Interestingly (and maybe surprisingly), for all the correlations presented above, every time that the leading term is algebraic, such off-diagonal term always leave it untouched.

Let us consider as a first example $C_{2,\gamma}^{\pm}(x,t)$, defined in Eq. (42) of Section 5.1. In this correlation, the leading term with the assumption of factorized initial state are the exponential terms in Eq. (44) coming from the massive sector. The massless sector provides only a correction to the truly asymptotic behavior which is reported in Eq. (45). The corresponding exact term from the Bogoliubov approach is reported in the appendix, cfr. Eq. (98). It is evident that the two are in general different: in fact, while (45) only has two free parameters (i.e., $\mathbb{K}_\pm$), in (98) we have four of them (i.e., $\Theta$ and $\{a_{ij}\}$ in (98), defined in Appendix C). Remarkably, they exhibit the same lightcone structure.

Nonetheless, as anticipated in the previous section, it turns out that in short time regime also power laws are correctly captured. This agreement does not come as a surprise, because, in the short-time regime (namely, before the first lightcone), the correlations reduce to the ones in the initial state. Anyhow, this obvious result comes from a nontrivial limit and was worth to test. Moreover, since this is the only regime where power-laws become leading (cf. (44)), the conclusion is that the leading term in both $x$ and $t$ is *always* correctly captured by our approach. Note that in the intermediate regimes $x$ and $t$ have to scale in the same way (by definition, $x/t$ must be finite and within the limits defining the corresponding regime). Therefore in this case, we have effectively just one independent variable, and the leading term is exponential. Coincidently, in the

specific case $K_1 = K_2$, the exponential decay in $x$ also vanishes in the prethermal regime (i.e., $2u_2 t > x > |u_1 - u_2|t$), and one can verify that the power law correction is correctly described as well.

The other correlation function of interest is the density-density one that we considered in Section 5.2. The effect of having $\gamma \neq 0$ in this correlation is to add a mixing term $\langle\!\langle \partial\theta_A \partial\theta_B \rangle\!\rangle_\gamma$ to Eq. (48). However, it is clear that this further term decay with the *same* power-law as the leading B/B term in Eq. (48): therefore the presence of $\gamma$ only changes the prefactor in front of the power-law decay.

Note that there is a main difference between the two examples discussed above. For the vertex operators, the correlations of $\theta_{A/B}$ appear at the exponent. As a consequence, the non-diagonal contribution from $\gamma \neq 0$ are *multiplicative* ones and therefore they just renormalize the power-laws. For the derivative operators, instead, the corrections in $\gamma$ are *additive*, and thus they do not change the power-law exponent, but just modify the non-universal prefactor.

## 5.4   Particular limits

In this short subsection, we analyze how our general quench simplifies when the velocities or the TLL parameters of the two species are the same.

### 5.4.1   Same velocities $u_1 = u_2$

When $u_1 = u_2$ (even for $K_1 \neq K_2$) in the post-quench Hamiltonian (1) $H_{u_1,K_1}$ and $H_{u_1,K_2}$ have the same spectrum. As a result, for any choice of the basis parametrized by $\nu$ in Eq. (21), the hamiltonian corresponds to two decoupled TLLs, as can be seen from (24). The initial state then selects a particular angle $\nu$ (cf. Eq. (20)) that ensures the decoupling between the massive and the massless degrees of freedom in the pre-quench Hamiltonian (3). In this case therefore, the initial state is *exactly* factorized in the two sectors and consequently the solution of non-equilibrium dynamics does not require any time rescaling (as a trivial consequence of the equality of two velocities). Hence, one can compute simply equal time correlation functions, and the light cone structure is largely simplified: this is evident, e.g., in (44) where one is left with two regimes only, corresponding to a unique lightcone.

Because of the perfect decoupling, the initial quench in the two TLLs induces a quench in the massive sector only, whereas the massless sector remains at equilibrium in the ground state. In fact the massless quench induced in the general case $u_1 \neq u_2$ results from an effective sudden change in the TLL parameters, but in this case they turn out to be equal. Specifically, one has $\Gamma/K_+ = 1$ if $u_1 = u_2$ (as can be checked from Eq. (10)). All previously reported results match this particular limit, but for some correlations in a singular manner requiring the restating of the short-distance cutoff.

### 5.4.2   Same TLL parameters $K_1 = K_2$

Contrarily to the limit of equal velocities, the one of equal TLL parameters does not bring major simplifications: the lightcone structure in (44), for example, remains since it is clearly related to the presence of different velocities only.

However, we note that in this case, in the correlation function (44), there is no exponential decay in space in the "prethermal" regime (namely $2u_2 t > x > |u_1 - u_2|t$) for $C_{2,0}^+$. Although there is still non-trivial time dependence in (44), the lack of exponential in $x$ can be interpreted as a prethermal "temperature" equal to zero. This last fact can be understood, at speculative level, by noting that in the prethermal regime the difference

in the velocities is small compared to the the considered spacetime scale ($x/t \gg |u_1 - u_2|$). Therefore, as $K_1 = K_2$, the total system is basically equivalent to two identical TLLs, and in that case the symmetric mode remains in its ground state [111]. In this regime the power law decay in space, coming from the massless sector, which becomes dominant, is correctly captured by our low energy approach.

# 6    Conclusions

In this work we have studied the quench dynamics of two coupled Tomonaga-Luttinger liquids from the off-critical to the critical conformal regime. This situation is relevant for several systems, including tunnel coupled condensates [75–83], the Hubbard model [8, 72], the Gaudin-Yang model [73, 74] or, more generally speaking, systems with two types of degrees of freedom.

We have shown that, for what concerns the large scale properties, this non-equilibrium dynamics decouples into two independent sectors, inducing an *effective* quench in each of them: one starting from a massive initial state and one from a massless one with an effective TLL parameter. Each of them can be studied by means of the known techniques reported in Sections 3.1 and 3.2. The equal-time correlations of the coupled system map to correlation functions at different times of two uncorrelated modes. We have also discussed that, while the leading term and the light cone structure are always well captured by our approach, this is not the case for subleading power law corrections, generated by the coupling between these two modes (i.e., $\gamma \neq 0$ in Eq. (28)).

Moreover, a direct inspection of the correlation functions shows that while for vertex operators the leading contribution is given by the exponential decay of the massive mode and the massless mode acts with subleading multiplicative power law corrections, for derivative operators the large scale properties are determined by the power law decay of the massless mode, while the massive part constitutes an additive correction here. Therefore while vertex operators mimic a thermal like behavior, derivative operators behave as at $T = 0$ or more generally in a GGE [140].

A final interesting remark is that for the quench induced in the massless sector, the product of the speed of sound and the Luttinger parameter before and after the quench is equal (i.e., $v\Gamma = u_B K_+$ in our notation). This fact suggests that the quench respects Galilean invariance [39].

This work paves the way to the study of quenches in systems consisting of more than a single TLL. We conclude by providing some future possible developments in this direction. The most natural extension of our calculation would be to apply our CFT approach to initial states in which the orthogonal modes correspond to two massive or two massless theories. To this aim, it is sufficient to change the parametrization of the initial squeezed state (28) with two eigenvalues with absolute values equal to one (massive case) or smaller than one (massless case).

Another generalization would be to consider a larger number of initially coupled TLLs. In this case, there are many different physical situations requiring different numbers of massive and massless modes. In particular, in the case where the TLLs correspond to many different tubes, it would be interesting to investigate how two-dimensional non-equilibrium physics emerges from the coupling of 1D systems, as done in a very different setup in [85].

Finally we mention possible connections with the works on conformal interfaces [141, 143–146], that we plan to investigate in the future [142]. Our framework, in fact, can be in principle reformulated in a full path integral fashion via the *unfolded picture* [143,

144], where the initial state (living in the tensor product of two CFTs, i.e., $\text{CFT}_1 \times \text{CFT}_2$) is mapped to an interface (connecting two spatial regions in a single CFT, namely $\text{CFT}_1 \cup \overline{\text{CFT}_2}$).

# Acknowledgements

We would like to thank Jérôme Dubail for useful discussions. This work is supported by "Investissements d'Avenir" LabEx PALM (ANR-10-LABX-0039-PALM), EquiDystant project (LF) and by the Swiss National Science Foundation under Division II (PR and TG). PC acknowledges support from ERC under Consolidator grant number 771536 (NEMO).

# A    Calculations in the massive sector

In the path integral approach in imaginary time developed in [33, 35], quantities as the one in Eq. (38) are mapped to correlation functions in a strip geometry with boundary conditions corresponding to conformally invariant boundary states. In (1+1)-dimensional BCFT, those are computed exploiting the transformations of correlation functions of (primary) operators under conformal maps. Let us consider, for example, two geometries in the complex plane with a boundary (say G1 and G2) with coordinate $w$ and $z$, related by the conformal map $w(z)$. Then the correlations of primary operators $\phi_i$ in the two geometries are related as

$$\left\langle \prod_i \phi_i(w_i, \bar{w}_i) \right\rangle_{\text{G1}} = \prod_i \left| w'(z_i) \right|^{h_i} \left| \bar{w}'(\bar{z}_i) \right|^{\bar{h}_i} \left\langle \prod_i \phi_i(z_i, \bar{z}_i) \right\rangle_{\text{G2}}, \tag{56}$$

with $h_i$ and $\bar{h}_i$ being the holomorphic and anti-holomorphic dimensions of $\phi_i$.

## A.1    Two-point function in the slab

To get the two-point function in (38), the object that we need to evaluate is a path integral over a slab (of width $W$) with the operators inserted, i.e.,

$$\langle V_{\alpha_1}(r, \sigma_1) V_{-\alpha_2}(r, \sigma_2) \rangle_{\text{slab}(W)} \tag{57}$$

with $V_\alpha = e^{i\alpha\theta}$ and $\theta$ is a bosonic field. Moreover $\sigma_i$ ($i = 1, 2$) are imaginary times, to be analitically continued to the values $\sigma_i \to it_i + W/2$ at the very end of the calculation.

Moving to complex coordinates (with points labelled by $w = r + i\sigma$), the correlation function of vertex operators $V_{\alpha_i}(w_i, \bar{w}_i)$ on the slab geometry is first mapped by a conformal transformation to the upper-half plane (UHP) (with coordinate $z$ s.t. $\text{Im}(z) > 0$). The two-point function in the UHP is related to a four-point function of *chiral* vertex operators $V_\alpha(z_i)$ on the complex plane, $z \in \mathbb{C}$ [147]. The details of the calculation can be found in [35]. The final result is

$$\langle V_{\alpha_1}(w_1, \bar{w}_1) V_{-\alpha_2}(w_2, \bar{w}_2) \rangle_{\text{slab}(W)} = \text{J} \times \langle V_{\alpha_1}(z_1, \bar{z}_1) V_{-\alpha_2}(z_2, \bar{z}_2) \rangle_{\text{UHP}} =$$

$$= \text{J} \times \langle V_{\alpha_1}(z_1) V_{-\alpha_2}(z_2) V_{-\alpha_1}(\bar{z}_1) V_{\alpha_2}(\bar{z}_2) \rangle_{\mathbb{C}} = \text{J} \times \left( \frac{|z_{1\bar{2}}|^2}{|z_{12}|^2} \right)^{2\sqrt{h_1 h_2}} \left( \frac{1}{|z_{1\bar{1}}|} \right)^{2h_1} \left( \frac{1}{|z_{2\bar{2}}|} \right)^{2h_2} \tag{58}$$

conveniently express in terms of

$$z_i = R_i e^{i\frac{\pi}{W}\sigma_i}, \quad |z_i'| = \frac{\pi}{W}R_i, \quad z_{i\bar{i}} = 2R_i \sin\gamma_i,$$
$$|z_{ij}|^2 = R_i^2 + R_j^2 - 2R_iR_j\cos(\gamma_i - \gamma_j),$$
$$|z_{i\bar{j}}|^2 = R_i^2 + R_j^2 - 2R_iR_j\cos(\gamma_i + \gamma_j), \tag{59}$$

where $J = (\pi/W)^{2(h_1+h_2)}$ denotes the Jacobian factor in (56), $h_i = \alpha_i^2/(8K_A)$ is the conformal weight of the chiral operator $V_{\alpha_i}(z)$, $\gamma_i = \pi\sigma_i/W$, $z_{ij} = |z_i - z_j|$, $z_{\bar{i}} = \bar{z}_i$, and $R_i = e^{\frac{\pi}{W}r_i}$ (in our case $R_1 = R_2$). Plugging into this expression the actual coordinates, we get

$$\langle V_{\alpha_1}(w_1, \bar{w}_1)V_{-\alpha_2}(w_2, \bar{w}_2)\rangle_{\text{slab}(W)} =$$

$$= \left(\frac{\pi}{W}\right)^{2(h_1+h_2)} \left(\frac{1 - \cos\left(\frac{\pi}{W}|\sigma_1 + \sigma_2|\right)}{1 - \cos\left(\frac{\pi}{W}|\sigma_1 - \sigma_2|\right)}\right)^{2\sqrt{h_1 h_2}} \left[2\sin\left(\frac{\pi}{W}\sigma_1\right)\right]^{-2h_1} \left[2\sin\left(\frac{\pi}{W}\sigma_2\right)\right]^{-2h_2}. \tag{60}$$

Finally, by analytically continuing $\sigma_i$ to real times and taking $t_i \gg W$, we obtain

$$\langle V_{\alpha_1}(r, t_1)V_{-\alpha_2}(r, t_2)\rangle = \left(\frac{\pi}{W}\right)^{2(h_1+h_2)} e^{\frac{2\pi}{W}\left(\sqrt{h_1 h_2}(|t_1+t_2|-|t_1-t_2|)-h_1|t_1|-h_2|t_2|\right)}. \tag{61}$$

Upon specifying the values of $\alpha_i$ ($i = 1, 2$), the above equation allows to access the massive component (38) of the one-point function $C_{1,0}^{\pm}(t)$ (cf. Eq. (37) in the main text). In particular, the difference between the symmetric and antisymmetric correlation boils down to the sign of $\alpha_2$. It is easy to realize that this is equivalent to consider a different sign in the corresponding time $t_2$ (cf. Eq (58)). Collecting all these observations, the correlation of interest read

$$\langle V_{\alpha_1}(r, t_1)V_{-\alpha_2}(r, \pm t_2)\rangle_{\text{slab}(W)} \sim e^{\frac{2\pi}{W}\left(\sqrt{h_1 h_2}(\pm|t_1+t_2|\mp|t_1-t_2|)-h_1|t_1|-h_2|t_2|\right)}. \tag{62}$$

## A.2   Four-point function on the slab

A similar calculation can be carried over for the the four-point function in the A-sector in Eq. (43). In the path integral formulation, the object of interest is

$$\langle V_{\alpha_1}(r, \sigma_1)V_{\alpha_2}(r, \sigma_2)V_{\alpha_1}(0, \sigma_1)V_{\alpha_2}(0, \sigma_2)\rangle_{\text{slab}(W)} \tag{63}$$

where all fields and variables are defined above. The calculation works in the exact same way, the only difference being the number of operator insertions

$$\langle V_{\alpha_1}(w_1, \bar{w}_1)V_{-\alpha_2}(w_2, \bar{w}_2)V_{-\alpha_1}(w_3, \bar{w}_3)V_{\alpha_2}(w_4, \bar{w}_4)\rangle_{\text{slab}(W)} =$$

$$= J \times \langle V_{\alpha_1}(z_1, \bar{z}_1)V_{-\alpha_2}(z_2, \bar{z}_2)V_{-\alpha_1}(z_3, \bar{z}_3)V_{\alpha_2}(z_4, \bar{z}_4)\rangle_{\text{UHP}}$$

$$= J \times \left(\frac{|z_{14}||z_{23}||z_{1\bar{2}}||z_{3\bar{4}}|}{|z_{12}||z_{34}||z_{1\bar{4}}||z_{2\bar{3}}|}\right)^{2\sqrt{h_1 h_2}} \left(\left|\frac{z_{1\bar{3}}}{z_{13}}\right|^2 \frac{1}{z_{1\bar{1}}z_{3\bar{3}}}\right)^{2h_1} \left(\left|\frac{z_{2\bar{4}}}{z_{24}}\right|^2 \frac{1}{z_{2\bar{2}}z_{4\bar{4}}}\right)^{2h_2}. \tag{64}$$

The relations (59) hold with now $i = \{1, 2, 3, 4\}$, and Eq. (63) corresponds to the special case $\gamma_1 = \gamma_3, \gamma_2 = \gamma_4, R_1 = R_2 \equiv R, R_3 = R_4 = 1$.

Upon analytic continuation to real times, and taking all scales much larger than $W$, we get to the expression

$$\langle V_{\alpha_1}(r, t_1)V_{\alpha_2}(r, t_2)V_{\alpha_1}(0, t_1)V_{\alpha_2}(0, t_2)\rangle$$

$$\sim \left[\frac{e^{\frac{2\pi}{W}(|t_1+t_2|)}}{e^{\frac{2\pi}{W}(|t_1-t_2|)}}\left(\frac{e^{\frac{2\pi}{W}r} + e^{\frac{2\pi}{W}|t_1-t_2|}}{e^{\frac{2\pi}{W}r} + e^{\frac{2\pi}{W}|t_1+t_2|}}\right)\right]^{2\sqrt{h_1 h_2}} e^{-\frac{2\pi}{W}r(h_1+h_2)}\left(1 + \frac{e^{\frac{\pi}{W}r}}{e^{\frac{2\pi}{W}|t_1|}}\right)^{2h_1}\left(1 + \frac{e^{\frac{\pi}{W}r}}{e^{\frac{2\pi}{W}|t_2|}}\right)^{2h_2}.$$

$$\tag{65}$$

For $C_{2,0}^{\pm}(t)$, everything simplifies to

$$\langle V_{\alpha_1}(r,t_1)V_{\alpha_2}(r,\pm t_2)V_{\alpha_1}(0,t_1)V_{\alpha_2}(0,\pm t_2)\rangle \sim$$

$$\sim \begin{cases} e^{\frac{4\pi}{W}\left[\pm\sqrt{h_1 h_2}(|t_1+t_2|-|t_1-t_2|)-h_1|t_1|-h_2|t_2|\right]} & r > 2t_1 \\ e^{\frac{4\pi}{W}\left[\pm\sqrt{h_1 h_2}(|t_1+t_2|-|t_1-t_2|)\right]}e^{-\frac{2\pi}{W}r(h_1-2h_2|t_2|)} & |t_1+t_2| < r < 2t_1 \\ e^{\frac{4\pi}{W}\left[\pm\sqrt{h_1 h_2}(r-|t_1-t_2|)\right]}e^{-\frac{2\pi}{W}r(h_1-2h_2|t_2|)} & 2t_2 < r < |t_1+t_2| \\ e^{\frac{4\pi}{W}\left[\pm\sqrt{h_1 h_2}(r-|t_1-t_2|)\right]}e^{-\frac{2\pi}{W}r(h_1+h_2)} & |t_1-t_2| < r < 2t_2 \\ e^{-\frac{2\pi}{W}r(h_1+h_2)} & r < |t_1-t_2|. \end{cases} \quad (66)$$

# B  Calculations in the massless sector

For computing the contribution from the massless sector $(B)$, we rely on the approach of [34], based on Bogoliubov transformations. In [34], the author focuses on two-point correlation functions at equal times. When the two sound velocities are different $(u_1 \neq u_2)$, we end up in correlators at different times that we provide in what follows.

## B.1  Two-point function at different times

Here we derive Eqs. (40) and (41) in the main text, that enter in the $B$-sector contribution to Eq. (37). This is just the two-point function for a quench $K_0 \to K_f$ in a TLL (below the unique sound velocity is set to 1), for which we use the notations introduced in Section 3.2. We consider $t_1 \neq t_2$ and, without losing generality, we take $t_1 > t_2$.

We compute the general correlation

$$\langle e^{i\alpha[\theta(x,t_1)-\theta(0,t_2)]}\rangle = e^{-\frac{\alpha^2}{2}\langle[\theta(x,t_1)-\theta(0,t_2)]^2\rangle}, \quad (67)$$

with $\alpha \in \mathbb{R}$ and, working in the Heisenberg picture, the expectation value is on the ground state of the Luttinger liquid hamiltonian with Luttinger parameter $K_0$.

The correlation in the exponent in Eq. (67) can be decomposed as

$$\langle[\theta(x,t_1)\pm\theta(0,t_2)]^2\rangle = \langle\theta(x,t_1)^2\rangle + \langle\theta(0,t_2)^2\rangle \pm 2\langle\theta(x,t_1)\theta(0,t_2)\rangle \quad (68)$$

where each of the terms above is a two-point function of $\theta$ at equal or different times. Then, using the following decomposition in modes for the field (in terms of the post-quench ladder operators $b_p$)

$$\theta(x,t) = \frac{i}{\sqrt{L}}\sum_{p\neq 0}e^{ipx}\sqrt{\frac{\pi}{2K_f|p|}}(b_p^\dagger(t)-b_{-p}(t)), \quad (69)$$

and taking the thermodynamic limit $(L \to \infty)$ we get

$$\langle\theta(x,t_1)\theta(0,t_2)\rangle = \frac{1}{2K_f}\int_0^\infty \frac{dp}{p}\cos(px)\left[\mathfrak{B}^\dagger U^\dagger(t_1)I_2 U(t_2)\mathfrak{B}\right]_{22} \quad (70)$$

where $\mathfrak{B}$ is the Bogoliubov matrix

$$\mathfrak{B} = \begin{pmatrix} \cosh\delta & -\sinh\delta \\ -\sinh\delta & \cosh\delta \end{pmatrix}, \qquad \delta = \frac{1}{2}\log(\frac{K_0}{K_f}), \quad (71)$$

and we further defined

$$I_2 = \begin{pmatrix} 1 & -1 \\ -1 & 1 \end{pmatrix}, \quad U(t) = \begin{pmatrix} e^{-i|p|t} & 0 \\ 0 & e^{i|p|t} \end{pmatrix}. \quad (72)$$

Finally, in Eq. (70) we denoted as $[\cdot]_{ij}$ the elements of a given matrix. Note that (70) is in general not real, however we will only be interested in real combinations of terms like in Eq. (68). For different times ($t_1 \neq t_2$) one finds

$$\left[\mathfrak{B}^\dagger U^\dagger(t_1) I_2 U(t_2) \mathfrak{B}\right]_{22} = \left(e^{ipt_2}\cosh\delta - e^{-ipt_2}\sinh\delta\right)\left(e^{-ipt_1}\cosh\delta - e^{ipt_1}\sinh\delta\right) \quad (73)$$

while, at equal times ($t_1 = t_2 \equiv t$), it simplifies to

$$\left[\mathfrak{B}^\dagger U^\dagger(t) I_2 U(t) \mathfrak{B}\right]_{22} = \cosh(2\delta) - \cos(2pt)\sinh(2\delta). \quad (74)$$

For $t = 0$, Eq. (74) simpifies to $K_f/K_0$, so that the correlations like (70) only depends on $K_0$ as they should. Eq. (68) then reads

$$\langle[\theta(x,t_1) \pm \theta(0,t_2)]^2\rangle = \frac{1}{2K_f}\int_0^\infty \frac{dp}{p}\left\{\mathbb{K}_+ - \frac{\mathbb{K}_-}{2}(\cos(2pt_1) + \cos(2pt_2))+\right.$$
$$\left. \pm\cos(px)\left[\mathbb{K}_+\cos(p|t_1-t_2|) - \mathbb{K}_-\cos(p|t_1+t_2|)\right]\right\}, \quad (75)$$

where we defined

$$\mathbb{K}_\pm = \frac{K_0}{K_f} \pm \frac{K_f}{K_0}. \quad (76)$$

Note that the leading term in (75) diverges as $\sim 1/p$, giving rise to a power decay in (67).

In the case of equal spatial points, Eq. (75) simplifies to

$$\langle[\theta(0,t_1) \pm \theta(0,t_2)]^2\rangle = \frac{1}{2K_f}\int_0^\infty \frac{dp}{p}\times$$
$$\times\left\{\mathbb{K}_+\left(1 \pm \cos(p(t_1-t_2))\right) - \mathbb{K}_-\left[\frac{1}{2}(\cos(2pt_1) + \cos(2pt_2)) \pm \cos(p(t_1+t_2))\right]\right\}. \quad (77)$$

For $K_0 = K_f$, we are computing a correlation function at equilibrium in the ground state. Accordingly, the expression above becomes time translational invariant (only the term involving the times difference survives).

## B.2 Four-point function at different times

Since the theory is quadratic, the calculation of higher point correlation functions can always be reduced to that of two-point functions. We will see it explicitly below in the case of the four-point function considered in the main text in Eq. (45). We start by noting that

$$\left\langle e^{i\alpha[(\theta(x,t_1)-\theta(0,t_1))\pm(\theta(x,t_2)-\theta(0,t_2))]}\right\rangle = e^{-\frac{\alpha^2}{2}\langle[(\theta(x,t_1)-\theta(0,t_1))+(\theta(x,t_2)-\theta(0,t_2))]^2\rangle} \quad (78)$$

which follows directly from Wick theorem. Then, we proceed by splitting the exponent in the rhs of (78) in three pieces as follows

$$\langle[(\theta(x,t_1) - \theta(0,t_1)) \pm (\theta(x,t_2) - \theta(0,t_2))]^2\rangle =$$
$$\langle[\theta(x,t_1) - \theta(0,t_1)]^2\rangle + \langle[\theta(x,t_2) - \theta(0,t_2)]^2\rangle$$
$$\pm 2\langle[\theta(x,t_1) - \theta(0,t_1)][\theta(x,t_2) - \theta(0,t_2)]\rangle. \quad (79)$$

This splitting is particularly convenient because each term is infared finite, so that no cutoff is needed at small $p$.

The first two terms in (79) are of the form (75) evaluated at equal times. Performing the integral (with an UV cutoff) we get

$$\langle [\theta(x,t) - \theta(0,t)]^2 \rangle = \frac{\mathbb{K}_+}{2K_f} \frac{1}{2} \log |x^2| - \frac{\mathbb{K}_-}{2K_f} \frac{1}{2} \log \left| 1 - \frac{x^2}{(2t)^2} \right|. \tag{80}$$

For the last term in (79), we find

$$2\langle [\theta(x,t_1) - \theta(0,t_1)] [\theta(x,t_2) - \theta(0,t_2)] \rangle =$$
$$= \frac{1}{K_f} \int_0^\infty \frac{dp}{p} (1 - \cos(px)) [\mathbb{K}_+ \cos(t_1 - t_2) - \mathbb{K}_- \cos(t_1 + t_2)]$$
$$= \left( \frac{\mathbb{K}_+}{K_f} \frac{1}{2} \log \left| 1 - \frac{x^2}{(t_1 - t_2)^2} \right| - \frac{\mathbb{K}_-}{K_f} \frac{1}{2} \log \left| 1 - \frac{x^2}{(t_1 + t_2)^2} \right| \right). \tag{81}$$

Putting everything together and performing trivial algebraic simplifications, we get

$$\langle [(\theta(x,t_1) - \theta(0,t_1)) \pm (\theta(x,t_2) - \theta(0,t_2))]^2 \rangle =$$
$$\log \left| x^2 \right|^{\frac{\mathbb{K}_+}{2K_f}} \left| 1 - \frac{x^2}{(t_1 - t_2)^2} \right|^{\pm \frac{\mathbb{K}_+}{2K_f}} \left| 1 - \frac{x^2}{(2t_1)^2} \right|^{-\frac{\mathbb{K}_-}{4K_f}} \left| 1 - \frac{x^2}{(2t_2)^2} \right|^{-\frac{\mathbb{K}_-}{4K_f}} \left| 1 - \frac{x^2}{(t_1 + t_2)^2} \right|^{\mp \frac{\mathbb{K}_-}{2K_f}}. \tag{82}$$

# C Calculations in the exact state: Bogoliubov approach

For comparison, we briefly sketch the calculations for the same correlation functions within the Bogoliubov approach. More details can be found in Ref. [116].

## C.1 One-point function of $\theta_\pm$

The one-point function in Eq. (36) can be written as

$$C_{1,\gamma}^\pm(t) = e^{-\langle\!\langle \theta_\pm^2(t) \rangle\!\rangle_\gamma} = e^{-\int dp \langle\!\langle \theta_{\pm,p}(t)\theta_{\pm,-p}(t) \rangle\!\rangle_\gamma}, \tag{83}$$

with

$$\langle\!\langle \theta_{\pm,p}(t)\theta_{\pm,-p}(t) \rangle\!\rangle_\gamma = \frac{1}{2} \sum_{i,j=1}^2 (-1)^{i+j} \langle\!\langle \theta_{i,p}(t)\theta_{j,-p}(t) \rangle\!\rangle_\gamma, \tag{84}$$

where used the decomposition

$$\theta_i(x,t) = \sum_p e^{-ipx} \theta_{i,p}(t), \tag{85}$$

and $(\alpha_{i,p} = \frac{\pi}{K_i|p|})$

$$\theta_{i,p}(t) = \cos(u_i|p|t)\theta_{i,p}(0) - \alpha_{i,p} \sin(u_i|p|t)n_{i,p}(0). \tag{86}$$

Hence, the expectation value in the exponent in (83) is

$$\langle\!\langle \theta_{i,p}(t)\theta_{j,-p}(t) \rangle\!\rangle_\gamma =$$
$$= \cos(u_i pt)\cos(u_j pt)\langle\!\langle \theta_{i,p}\theta_{j,-p} \rangle\!\rangle_\gamma + \sin(u_i pt)\sin(u_j pt)\alpha_{i,p}\alpha_{j,-p}\langle\!\langle n_{i,p}n_{j,-p} \rangle\!\rangle_\gamma. \tag{87}$$

The small $p$ expansion of (87) reads

$$\langle\!\langle\theta_{i,p}(t)\theta_{j,-p}(t)\rangle\!\rangle_\gamma = \frac{\mathcal{A}_{ij}(p)}{p^2} + \frac{\mathcal{B}_{ij}(p)}{p} + O(0), \tag{88}$$

with $\mathcal{A}_{ij}(p)$ and $\mathcal{B}_{ij}(p)$ regular for $p \to 0$. The leading contribution in (83) comes from the term $\propto 1/p^2$. This contribution was explicitly computed in [116] and gives an exponential decay in (83).

Now, we consider the next-to-leading contribution $\propto 1/p$ in (88). The explicit expression for $\mathcal{B}$ is

$$\mathcal{B}_{ij}(p) = \Theta_{ij}\cos(u_ipt)\cos(u_jpt) + \Pi_{ij}\sin(u_ipt)\sin(u_jpt) \tag{89}$$

where we defined

$$\Theta_{ij} \equiv \lim_{p\to0} p\langle\!\langle\theta_{i,p}\theta_{j,-p}\rangle\!\rangle_\gamma = \frac{\pi}{4\Gamma} \equiv \Theta\,, \tag{90}$$

and

$$\Pi_{ij} = \lim_{p\to0} p\,\alpha_{i,p}\alpha_{j,-p}\langle\!\langle n_{i,p}(t)n_{j,-p}\rangle\!\rangle_\gamma = \Theta\,a_{ij}^2. \tag{91}$$

Integration over momentum of Eq. (89) gives ($y = pt$)

$$\int \frac{dp}{2\pi}\frac{\mathcal{B}_{ij}(p)}{p} = \frac{\Theta}{2\pi}\left\{\frac{1+a_{ij}^2}{2}\int dy\frac{\cos(|u_i-u_j|y)}{y} + \frac{1-a_{ij}^2}{2}\int dy\frac{\cos(|u_i+u_j|y)}{y}\right\}. \tag{92}$$

Then, using Eq. (84), and the expansion

$$\langle\!\langle\theta_{\pm}(t)\theta_{\pm}(t)\rangle\!\rangle_\gamma = \int \frac{dp}{2\pi}\left(\frac{\mathcal{A}_{\pm}(p)}{p^2} + \frac{\mathcal{B}_{\pm}(p)}{p} + O(0)\right), \tag{93}$$

we find

$$\int \frac{dp}{2\pi}\frac{\mathcal{B}_{\pm}(p)}{p} = \frac{\Theta}{2\pi}\left\{\frac{a_{11}^2+a_{22}^2}{2}\int\frac{dz}{z}(1-\cos z) + \int\frac{dz}{z}(1+(1\pm2)\cos z)\right\} \tag{94}$$

The above integral is convergent in one case $(-)$, while diverges in the other $(+)$ ones (due to the infrared behavior). Since it appear in the exponent for $C_{1,\gamma}^{\pm}$ (cfr. (83)), it implies an algebraic decay at large $t$ for that $C_{1,\gamma}^{-}$, and gives $C_{1,\gamma}^{+} = 0$.

## C.2  Two-point function of $\theta_{\pm}$

We can similarly derive the two-point function (42), i.e.

$$C_{2,\gamma}^{\pm}(x,t) = e^{-\langle\!\langle[\theta_{\pm}(x,t)-\theta_{\pm}(0,t)]^2\rangle\!\rangle_\gamma}\,. \tag{95}$$

Let us start by reintroducing the space dependence in (93) as follows

$$\langle\!\langle\theta_{\pm}(x,t)\theta_{\pm}(y,t)\rangle\!\rangle_\gamma = \int_0^\infty \frac{dp}{\pi}e^{ip(x-y)}\left(\frac{\mathcal{A}_{\pm}(p)}{p^2} + \frac{\mathcal{B}_{\pm}(p)}{p} + O(0)\right), \tag{96}$$

We are interested in the term in (96) whose integrand is $\propto 1/p$. Using (92), the latter reads

$$\int_0^\infty \frac{dp}{\pi}e^{ip(x-y)}\frac{\mathcal{B}_{\pm}(p)}{p} = \frac{\Theta}{2\pi}\Big[\frac{(1+a_{11}^2)+(1+a_{22}^2)}{2}\int\frac{dp}{p}e^{ip(x-y)}+$$

$$+\frac{(1-a_{11}^2)}{2}\int\frac{dp}{p}e^{ip(x-y)}\cos(2u_1pt) + \frac{(1-a_{22}^2)}{2}\int\frac{dp}{p}e^{ip(x-y)}\cos(2u_1pt)$$

$$\pm(1+a_{12}^2)\int\frac{dp}{p}e^{ip(x-y)}\cos(|u_1-u_2|pt) \pm (1-a_{12}^2)\int\frac{dp}{p}e^{ip(x-y)}\cos(|u_1+u_2|pt)\Big] \tag{97}$$

Finally, from the above expression, we get for $\langle\!\langle[\theta_\pm(x,t)-\theta_\pm(0,t)]^2\rangle\!\rangle_\gamma$ a contribution of the form

$$\frac{\Theta}{2\pi}\log|x^2|^{\frac{(2+a_{11}^2+a_{22}^2)}{2}}\left|1-\frac{x^2}{(2u_1t)^2}\right|^{\frac{(1-a_{11}^2)}{2}}\left|1-\frac{x^2}{(2u_2t)^2}\right|^{\frac{(1-a_{22}^2)}{2}}\left|1-\frac{x^2}{(|u_1-u_2|t)^2}\right|^{\pm(1+a_{12}^2)}\left|1-\frac{x^2}{(|u_1+u_2|t)^2}\right|^{\pm(1-a_1^2}$$

Plugged in (95), this is the final result.

# D   Calculations in the exact state: Coherent states

Some of the calculations in the main text are more easily done in the coherent states basis (in a path integral fashion), that we now review. To begin with, we consider a simple squeezed state of the form

$$|\psi\rangle = \prod_{p>0}e^{\mathsf{W}_p b_p^\dagger b_{-p}^\dagger}|0\rangle, \tag{99}$$

with $\mathsf{W}_p \in \mathbb{C}$. Let us define the coherent states $|z_p\rangle$ as follows

$$b_p|z_p\rangle = z_p|z_p\rangle, \quad |z_p\rangle = e^{z_p b_p^\dagger - z_p^* b_p}|0\rangle, \tag{100}$$

$$\langle z_p|w_p\rangle = e^{-\frac{1}{2}(|z_p|^2+|w_p|^2-2z_p^* w_p)}, \quad \mathbb{I} = \int\prod_p\frac{dz_p d\bar{z}_p}{\pi}|z\rangle\langle z| \tag{101}$$

where $b_p^{(\dagger)}$ are operators, $z_p, w_p \in \mathbb{C}$, and $|z\rangle = \otimes_p|z_p\rangle$. The norm of $|\psi\rangle$

$$\langle\psi|\psi\rangle = \int\prod_p\frac{dz_p d\bar{z}_p}{\pi}\langle\psi|z\rangle\langle z|\psi\rangle, \tag{102}$$

is computed as follows. Using the definitions in (100), we find

$$\langle\psi|z\rangle = \prod_{p>0}\langle 0|e^{\mathsf{W}_p b_p b_{-p}}|z_p,z_{-p}\rangle = \prod_{p>0}e^{\mathsf{W}_p z_p z_{-p}}e^{-\frac{1}{2}(|z_p|^2+|z_{-p}|^2)}. \tag{103}$$

Moreover, using also that $\langle z|\psi\rangle = \langle\psi|z\rangle^*$, the norm of $|\psi\rangle$ is written as a gaussian integral, which can be computed explictly

$$\langle\psi|\psi\rangle = \prod_{p>0}\frac{1}{1-\mathsf{W}_p^2} \equiv \mathsf{N}^2. \tag{104}$$

Moving to correlation functions, and taking into account the above normalization, we similarly find

$$\langle\frac{\psi}{\mathsf{N}}|b_q b_q^\dagger|\frac{\psi}{\mathsf{N}}\rangle = \frac{1}{1-\mathsf{W}_q^2}, \quad \langle\frac{\psi}{\mathsf{N}}|b_q^\dagger b_q|\frac{\psi}{\mathsf{N}}\rangle = \frac{\mathsf{W}_q^2}{1-\mathsf{W}_q^2}, \tag{105}$$

where we used the commutation relations to make $b_p$ act on $|z\rangle$.

We then consider the squeezed state of interest in this work, namely of the form

$$|\psi\rangle = \prod_{p>0}e^{(b_{A,p}^\dagger,b_{B,p}^\dagger)\mathbb{W}_p\begin{pmatrix}b_{A,-p}^\dagger\\b_{B,-p}^\dagger\end{pmatrix}}|0\rangle, \qquad \mathbb{W}_p = \begin{pmatrix}w_{AA} & w_{AB}\\w_{AB} & w_{BB}\end{pmatrix} \tag{106}$$

where in $w_{ab} = w_{ab}^p$ $(a, b \in \{A, S\})$ the $p$-dependence is implicit, and, for simplicity, we assumed them to be real. First we (re)define the coherent states as

$$|z\rangle = \bigotimes_p |z_{A,p}\rangle \otimes |z_{B,p}\rangle, \quad b_{A/B,p}|z_{A/B,p}\rangle = z_{A/B,p}|z_{A/B,p}\rangle. \tag{107}$$

and we start again from computing the norm of $|\psi\rangle$. Repeating the same steps above, we find

$$\langle\psi|z\rangle = \prod_{p>0} e^{(z_{A,p}, z_{B,p})\mathbb{W}_p \begin{pmatrix} z_{A,-p} \\ z_{B,-p} \end{pmatrix}} e^{-\frac{1}{2}(|z_{A,p}|^2 + |z_{A,-p}|^2 + |z_{B,p}|^2 + |z_{B,-p}|^2)}, \tag{108}$$

and $\langle z|\psi\rangle = \langle\psi|z\rangle^*$. Using this, the norm of the state (106) can be put in the form

$$\prod_{p>0} \int \frac{d\hat{Z}_p}{\pi^4} e^{\hat{Z}_p^T \hat{\mathbb{M}}_p \hat{Z}_p}, \quad \hat{\mathbb{M}}_p = \begin{pmatrix} \begin{pmatrix} & \mathbb{W}_p \\ \mathbb{W}_p & \end{pmatrix} & -\mathbb{I}_4 \\ -\mathbb{I}_4 & \begin{pmatrix} & \mathbb{W}_p \\ \mathbb{W}_p & \end{pmatrix} \end{pmatrix} \tag{109}$$

where we defined the vector $\hat{Z}_p = (z_{A,p}, z_{B,p}, z_{A,-p}, z_{B,-p}, z_{A,p}^*, z_{B,p}^*, z_{A,-p}^*, z_{B,-p}^*)$, $\mathbb{I}_4$ is a $4 \times 4$ identity matrix, and $\hat{\mathbb{M}}_p$ results in a $8 \times 8$ symmetric matrix. Expoiting its gaussian nature, the above integral can be evaluated analytically to get

$$\mathcal{N}^2 \equiv \langle\psi|\psi\rangle = \prod_{p>0} \frac{1}{\sqrt{\det \hat{\mathbb{M}}_p}}$$

$$= \prod_{p>0} \left[ \left( w_{AA} + w_{BB} + w_{AA}w_{BB} + w_{AB}^2 - 1 \right) \left( w_{AA} + w_{BB} + w_{AA}w_{BB} - w_{AB}^2 + 1 \right) \right]^{-1}. \tag{110}$$

Similarly, correlation functions can be evaluated making use of the property of gaussian integral

$$\int d\hat{Z} e^{\hat{Z}^T \hat{\mathbb{M}} \hat{Z}} f(\hat{Z}) = \sqrt{\frac{\pi^n}{\det \hat{\mathbb{M}}_p}} \left( e^{-\sum_{ij} \hat{\mathbb{M}}_{ij}^{-1} \partial_i \partial_j} \right) f(\hat{Z})|_{\hat{Z}=0}. \tag{111}$$

For example, with the definitions above,

$$\langle\psi|b_{A,-q} b_{A,q}^\dagger|\psi\rangle =$$

$$\prod_{p\neq q} \frac{1}{\sqrt{\det \hat{\mathbb{M}}_p}} \frac{1}{\sqrt{\det \hat{\mathbb{M}}_q}} \left( e^{-\sum_{ij} \hat{\mathbb{M}}_{ij}^{-1} \partial_i \partial_j} \right) \hat{Z}_{q,1} \hat{Z}_{q,3}|_{\hat{Z}=0} = \prod_{p\neq q} \frac{1}{\sqrt{\det \hat{\mathbb{M}}_p}} \left( \frac{-2\hat{\mathbb{M}}_{q,13}^{-1}}{\sqrt{\det \hat{\mathbb{M}}_q}} \right). \tag{112}$$

All the two-point functions of $b_{A/B,p}^{(\dagger)}$ can be collected in the following $4 \times 4$ matrix

$$\mathbb{B}_p = \langle \begin{pmatrix} b_{A,p}^\dagger \\ b_{A,-p} \\ b_{B,p}^\dagger \\ b_{B,-p} \end{pmatrix} \begin{pmatrix} b_{A,p} & b_{A,-p}^\dagger & b_{B,p} & b_{B,-p}^\dagger \end{pmatrix} \rangle = \begin{pmatrix} m_{15}-1 & m_{13} & m_{25} & m_{23} \\ m_{13} & m_{15} & m_{23} & m_{25} \\ m_{25} & m_{23} & m_{26}-1 & m_{24} \\ m_{23} & m_{25} & m_{24} & m_{26} \end{pmatrix} \tag{113}$$

where $m_{kl} = -2\hat{\mathbb{M}}_{p,kl}^{-1}$, and we exploited the symmetries of $\mathbb{B}_p$. Expectation values are understood on the normalized state $|\psi\rangle/\mathcal{N}$. In particular, we want to consider the $O(p^0)$

of $\mathbb{B}_p$, for the squeezed state in (106) with $\mathbb{W}_p = W_p$ (cfr. Eq. (28) in the main text), so that expectation values are given by $\langle\!\langle \cdot \rangle\!\rangle_\gamma$.

Using the following definitions (analogous to (90) and (91))

$$\Theta_{a,b} \equiv \lim_{p\to 0} p \langle\!\langle \theta_{a,p}\theta_{b,-p} \rangle\!\rangle_\gamma, \qquad \Pi_{a,b} = \lim_{p\to 0} p\, \alpha_{a,p}\alpha_{b,-p} \langle\!\langle n_{a,p}(t)n_{b,-p} \rangle\!\rangle_\gamma, \quad a,b \in \{A, B\}, \quad (114)$$

and by expanding $\theta_{A/B,p}$ and $n_{A/B,p}$ in terms of $b^\dagger_{A/B,p}$, one can check that

$$\Theta_{AA} = \Theta_{AB} = 0, \quad \Theta_{BB} = \frac{\pi}{2K_B}\frac{\cos 2\varphi - 1}{\cos 2\varphi + 1} \qquad (115)$$

namely they are independent on the value of $\gamma$. This is not the case for $\Pi_{ab}$, in which case one finds

$$\begin{aligned}
\Pi_{AA} &= \frac{\pi}{2K_A}\left(\frac{\gamma^2}{u_A^2\tau_A^2}\frac{2}{\cos 2\varphi - 1} - 1\right), \\
\Pi_{AB} &= \frac{\pi}{2\sqrt{K_A K_B}}\left(\frac{\gamma}{u_A\tau_A}\frac{2}{\cos 2\varphi - 1}\right), \\
\Pi_{BB} &= \frac{\pi}{2K_B}\left(\frac{1 + \cos 2\varphi}{1 - \cos 2\varphi}\right).
\end{aligned} \qquad (116)$$

Finally, note that $\tau_B$ never enters in the above expressions.

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
