# Peer review of "Quenches in initially coupled Tomonaga-Luttinger Liquids: a conformal field theory approach"

_SciPost Physics_

## Round 1 · Referee Report · Anonymous (Referee 1) · 2021-5-1

Strengths

1- Well structured presentation with detailed overview of available methods, results in main text and technical calculations in appendices. 2- Good account of relevant literature. 3- Main finding: reduction of massive-to-massless quench in two coupled Luttinger liquids to two decoupled quenches (one massive + one massless) allowing use of earlier results.

Weaknesses

1- Motivation, objectives, new results not explained in sufficient clarity/detail in the introduction: which results or methods are new/distinct compared to earlier Ref. [116]? 2- Effects of off-diagonal corrections not investigated in detail. 3- Absence of illustration figures and plots.

Report

The manuscript presents an analysis of quench dynamics in systems described by two coupled Luttinger liquids from the non-critical to the critical phase, which is a subject relevant to a number of significant experimental and theoretical applications. Based on a reasonable ansatz for the form of the initial state, the qualitative behavior of correlations of different types of observables is studied in detail. In particular, the large time asymptotics of vertex operators and derivative fields is derived and classified as following exponential or power-law scaling with respect to the time and distances between the points. The method used in these works relies on the approximate factorization of the initial state that reduces the problem to a combination of a massive and a massless quench that have already been analyzed in the relevant literature. The present work is an extension of that of Ref. [116] on the same problem, including methodological simplifications and additional results.

Even though it is not clearly stated which of the results presented here are new in comparison with the earlier work of Ref. [116], the main findings are the reduction of the quench problem to that of two decoupled sectors, one massive and one massless quench, which allows the derivation of the asymptotics of correlation functions using earlier methods. There are also new results on the asymptotics of derivative field correlations (Sec. 5.2). The presentation is well-structured, presenting the more technical parts of the calculation in appendices, and the introduction includes a good overview of the subject and account of earlier literature. Nevertheless, despite its length, a sufficiently detailed explanation of the motivation, objectives, and summary of results or their significance is missing from the introduction.

Given the above remarks, this submission does not clearly meet the criteria of SciPost Physics, but does meet those of SciPost Physics Core, in which I would recommend publication after the following requested changes.

Requested changes

1- The authors should explain in the introduction in what ways the present work goes beyond Ref. [116]: they should motivate and state the objectives of the present analysis in comparison with those of the previous one in a more detailed way and should also provide a summary of the new results.

2- In Eq. (28) it is argued that at next-to-leading order in a small momentum expansion the initial state receives corrections that are linear in momentum, with one of the coefficients being proportional to the inverse initial mass, as expected by RG considerations, however, these coefficients are not given explicitly. It is important to calculate the parameters $\tau_A, \tau_B$ and $\gamma$ exactly for the quench from (3) to (1), verify the agreement with the RG predictions regarding the functional dependence on the original parameters, and evaluate the relevance of the off-diagonal contributions to practical applications.

3- Even though a discussion of off-diagonal corrections to the initial state is presented in Sec. 5.3, it is not conclusively clarified what their effects are on the asymptotics of the observables studied here, if these corrections can be ignored or if they would result in some significant or noticeable deviation.

4- Unlike in CFT, in the Luttinger liquid description of the quench of Ref. [34] the parameters of the model are momentum dependent functions, which is standard in Luttinger liquid theory. This dependence can be ignored at equilibrium as it corresponds to irrelevant (higher order derivative) perturbations, resulting in the simple free boson CFT model, however this is not necessarily valid away from equilibrium. In fact such irrelevant perturbations may be expected to play an increasingly important role at large times. How would the asymptotic results presented here be modified by such considerations?

5- There is some minor ambiguity over what is meant by leading order asymptotics and in which regimes the reported results are expected to hold. Power-law multiplicative corrections to exponential scaling may not be considered as subleading corrections by all readers. Also, in the case of power-law asymptotics where there is a cancellation of the exponents, the leading asymptotics is controlled by what is considered here as subleading corrections. Lastly, it should be clarified in what regime the results are expected to hold in applications, i.e. how far from the lines separating the different space-time regions (at a finite distance or at distances increasing with time?).

6- Eq. (98) is chopped.

7- There is a rather large number of typos and English errors, most of which can be corrected by an automated language check.

  • validity: top
  • significance: good
  • originality: good
  • clarity: high
  • formatting: good
  • grammar: reasonable

Author:  Paola Ruggiero  on 2021-07-05  [id 1546]

(in reply to Report 1 on 2021-05-01)

We thank again the referee for the detailed report. Below we reply to all the raised points.

1- In order to emphasise motivation, objectives, and main results we added a subsection at the end of introduction (now Section 1.1) with this specific aim. In this new version, it should be clear that the specific solution of the quadratic quench in Ref. [116] is used to understand how to generically encode desired properties of the initial state, using a squeezed form with a specific low energy expansion of the matrix W_p (cf. Eq.(1)). In particular we show that a crucial information is encoded in its eigenvalues: an eigenvalue equal to one is related to a massive mode, while an eigenvalue smaller than one to a massless mode. The generality of our approach lies in that such a state represents an effective description of many possible initial states, including the physically relevant case of the ground state of two LLs with a generic coupling between them (leading to a massive and a massless mode). Indeed, it is clear that Eq. (1) reproduces the leading order of equilibrium correlation functions at large distances (tau_0 is an effective correlation length that can be adjusted to match the initial mass), and is here assumed to be also a good starting point from the out-of-equilibrium problem.

2-The explicit form of these coefficients can be straightforwardly worked out for the specific quench mentioned by the referee, but it is a complicated form of the initial parameters and is definitely not very enlightening. As stated in (1-), the main point of our approach is to show that the correlations functions at leading order are independent of the values of \gamma and \tau_B, that is the reason why we decided not to report their values. The parameter \tau_A takes instead the simple form tau_A=1/m_0, already reported in Eq. (38), with m_0 given in that case below Eq. (11) in that specific case.

3- Our claim is that if there is an exponential decay, this is correctly captured by the factorized approximation \gamma=0 up to power-law (multiplicative) corrections. If the leading term is a power law, in the examples considered, this is also correctly described. Whether such corrections “can be ignored” depends on the particular question one aims at answering.

4- While we clearly motivate that our initial state can be taken as an effective description of two LLs coupled by a generic RG relevant term in the sense that it reproduces the leading order of equilibrium correlation functions, we also clearly state that the same state being a good starting point from the out-of-equilibrium problem is an assumption. In general this is not guaranteed already for a single theory as, indeed, subleading corrections might become dominant at late times. However, it is natural to expect our results to be valid at least in a given time window. Whether at some later time our assumption is no longer valid is an open issue (that for a single theory is currently under investigation and we mentioned the relevant literature), that is beyond the aim of this paper.

5- In the text we now use the expression “leading behaviour”. The exact meaning of that is clarified in several point in the text, in order to avoid such “minor ambiguity”, and ultimately, in Section 5.3, which deals exactly with such corrections.

6- We corrected this minor point.

7- We corrected several typos as well.

Author:  Paola Ruggiero  on 2021-05-18  [id 1435]

(in reply to Report 1 on 2021-05-01)
Category:
remark
reply to objection

We thank the anonymous referee for his/her detailed report, comments and suggestions, and in particular for pointing out that readers could wonder what is the novelty of our results, as compared with the ones of Ref.[116], and their motivation. We did explain these point in the introduction but probably in a too rapid way. Needless to say we will strongly emphasize and further clarify these points in the introduction, when we will be asked to provide a revised of the manuscript.

Meanwhile let us just give a brief explanation on this important point asked by the referee (we will of course address the full detailed referee report separately together with the revision of the manuscript):

First of all, as stated (too briefly) in the introduction, our analysis is a generalization of [116] in that, while there a specific quench was considered (namely the initial state was the ground state of a massive free theory), here results are given for a wide class of states (those of the form in Eq.(28)), here argued to be general due to their RG interpretation.

Moreover, the approximate factorization we unveil is, to the best of our knowledge, new in this setting of coupled unequal LL, and is a highly non-trivial aspect of our analysis. Indeed, this makes the role of different terms in correlation functions transparent, allowing to attribute each of them to a massive or a massless mode, which, crucially, are independent at leading order.
The same factorization is what ultimately allows to compute new correlation functions, as the ones in Sec. 5.3, something that we were not able to access with the method in [116], where no factorization was identified.

---

## Round 1 · Referee Report · Anonymous (Referee 2) · 2021-6-4

Strengths

1)Analytical predictions for leading asymptotic in space/time of various correlation functions

2)Detailed work with several appendices with technical details

3)Long introductions with detailed account of literature

Weaknesses

1) Hard to grasp what the precise aim of this work actually is, within the scope of TLLs theory and beyond (applications to microscopic models such as the Hubbard model, referred to in the abstract and conclusion)

2)Presentation of the results could be improved

3)Relation with Ref 116 should be clarified better

Report

The focus of this work is to compute the quench dynamics of two initially coupled Tomonaga Lutting Liquids (TLLs) characterized by different parameters (sound velocity u, Luttinger parameter K) which then evolve independently, i.e. after switching off their mutual coupling.

In particular the authors aim at getting analytically the leading(next to leading) asymptotic in space/time of various quantities (one and two point functions, density/current correlation functions) using (i) a suitable writing of the initial state in terms of a set of modes whose dynamics can be computed easily, (ii) a suitable rescaling of time (to account for different sound velocities) (iii) an expansion in low momentum for the correlation matrix of the initial state.

The main result is that the problem reduces to computing quenches in two independent TLLs problems with the price to pay being that single time-observables become two-times quantities due to the rescaling. This allows the authors to get analytical results for several quantities.

I think this is a nice work, which however could benefit from some revision from the authors mainly concerning the clarity of their presentation, their main aim and the relation with previous works in the literature.

I attach few points below.

Once this is done I am happy to recommend this work for publication.

Requested changes

1) The introduction is very long and detailed, yet the space left to discuss the aim of the work and the relation to Ref 116 is rather limited. I suggest the authors to clarify their goal and how the present work differs from Ref 116 already in the introduction.

2)Section 2 is an important one - I would suggest the authors to clarify immediately that their focus is on initial states of the form of Eq.11. In this respect it is not clear to me what kind of initial states other than the ground state of Eq. 3 can be parametrized in this form. For example, the case of two coupled TLLs beyond the harmonic approximation (i.e. with a backscattering term in the phase difference) could also be written in this form? If so, in which regime? The authors refer often to "RG ideas" but being a bit more precise would help the reader.

3)This points to a more general issue: the authors refer often to Eq.3 but their discussion seems to suggest that also more general initial states can be treated with their approach. Can the authors be more explicit on this point, i.e. provide further examples? They mention in the abstract and conclusion the relevance of this work for the Hubbard model, but it is not clear to me how this is so - spin and charge separate at low energy but in general there will be backscattering terms in both sectors, how exactly these can be incorporated in an ansatz like Eq. 11? If so, in which regime? Without these clarifications the reader is left with the impression that the approach is suitable only for quadratic problems (before/after the quench), which however can also be solved with different methods (i.e. Ref 116)

4) I wonder if section 4 could not be better moved in the appendix, to avoid pushing the start of the original results (section 5) further below in the paper.

5)The discussion of the next to leading term in the low momentum expansion is presented first in Section 4.1.2 then in Section 5.3. I wonder if these discussions could not be merged and presented at a later stage, since -as far as I can tell - the main results are obtained without including the coupling between A/B modes.

6)A more elaborated comparison with the Bogolubov approach of Ref 116 could be useful for the reader. At the moment I could not find much about it in Section 5.3. The authors refer to several interesting results coming out from that work, it would be useful to mention more clearly what can be reproduced with their "low energy" approach and what cannot.

7)More of a physics question: what is the effect of having two different TLLs (with different parameters) in the initial and final Hamiltonian? On a technical level it is somehow clear (rescaling of time, mapping from one point to two points functions) but on a physical level (i.e.decay of correlation functions?) is there something new happening that depends on how different these two TLLs are? This might be discussed at somepoint in the manuscript, so my apologies if I missed it, but if there's something interesting there it would be nice to highlight it more.

  • validity: top
  • significance: good
  • originality: high
  • clarity: ok
  • formatting: good
  • grammar: good

Author:  Paola Ruggiero  on 2021-07-05  [id 1547]

(in reply to Report 2 on 2021-06-04)

We thank the referee for the comments and suggestions, that we tried to address as completely as possible . Below a point-by-point answer.

1) We now added a subsection (Sec. 1.1) exactly with this aim. As we already stated in the reply to Referee 1, it should now be clear that the specific solution of the quadratic quench in Ref. [116] is used to understand how to generically encode desired properties of the initial state, using a squeezed form with a specific low energy expansion of the matrix W_p (cf. Eq.(1)), with an eigenvalue equal to one being associated to a massive mode, and an eigenvalue smaller than one to a massless mode. Such a state can then be taken as an effective description of two LLs with a generic coupling, in the sense that it reproduces the leading order of equilibrium correlation functions.

2) These points are now already clarified at the beginning in the new Section 1.1 and further expanded in Section 2. The general idea is that, the addition to the hamiltonian of two LLs of a coupling term that opens a gap in the theory gives rise – at equilibrium – to exponentially decaying correlations (related with an effective mass), plus possible powerlaw corrections (multiplicative and/or additive ones). Such behaviour can be effectively reproduced by a squeezed state as in Eq. (1) with the above mentioned low-energy expansion (explicitly given in Eq. (27)). Crucially, such a state can always be though as the ground state of two TTLs with a quadratic coupling with mass given by the effective mass. The question of the time window in which such a state reproduces the exact out-of-equilibrium dynamics is an important open question beyond the aim of the paper, and in our work is an assumption, as clearly stated (for a single theory the question is currently under intense investigation and we mentioned the relevant literature).

3) The answer to question 2) above should clarify this point as well.

4) We thought about that, but we feel that Section 4 is really the core of the method, allowing for the factorization of correlation functions in Section 5, so we prefer to keep it in the main text.

5) The goal of Section 4.1.2 is to give the form of a generic initial state at O(p) and just point out that there is a complication which occurs because of the off-diagonal linear terms. Then the discussion about their role is done carefully in Section 5.3.

6) This point is now already clarified in Section 1.1, where we clearly state that we can on one side recover the results for the correlations of vertex operators already obtained in Ref. [116], but more than that we can easily obtain the correlations of density and currents. Moreover, differently from the Bogoliubov approach, this method allows us to clearly see the underlying structure of correlations functions, with separate contributions from massive and massless mode.

7)The rich phenomenology of the two different initially coupled TLLs is mentioned in the introduction: (i) the emergence of multiple lightcones, separating different decaying regimes; (ii) a prethermal regime eventually decaying into a quasi-thermal one; (iii) non-trivial effects of a non-zero temperature in the initial state. However, this more “physical” part is extensively discussed in Ref. [116] and we do not want to repeat too much ourselves (even to distinguish the result in [116] from the ones here as both referees asked).

---

## Editorial Decision

resubmitted